



# Bioaerosols in the Amazon rain forest: Temporal variations and vertical profiles of Eukarya, Bacteria and Archaea

Maria Prass[1], Meinrat O. Andreae[2,3], Alessandro C. de Araùjo[4], Paulo Artaxo[5], Florian Ditas[1,a], Wolfgang Elbert[1], Marco Aurélio Franco[1,5], Isabella Hrabe de Angelis[1], Jürgen Kesselmeier[1,2] , Thomas Klimach[1], Leslie Ann Kremper[1], Eckhard Thines[6,7], David Walter[1], Jens Weber[1], Bettina Weber[1,8], Bernhard M. Fuchs[9], Ulrich Pöschl[1], and Christopher Pöhlker[1]

1 Multiphase Chemistry Department, Max Planck Institute for Chemistry, 55128 Mainz, Germany

2 Biogeochemistry Department, Max Planck Institute for Chemistry, 55128 Mainz, Germany

3 Scripps Institution of Oceanography, University of California San Diego, La Jolla, CA 92083, USA

4 Empresa Brasileira de Pesquisa Agropecuária (EMBRAPA), Belém, PA, Brazil

5 Institute of Physics, University of São Paulo, São Paulo 05508-900, Brazil

6 Institute for Microbiology and Wine Research, Johannes Gutenberg University Mainz, 55128 Mainz, Germany

7 Institute of Molecular Physiology, Johannes Gutenberg University, 55128 Mainz, Germany

8 Department of Biology, University of Graz, Holteigasse 6, 8010, Graz, Austria

9 Department of Molecular Ecology, Max Planck Institute for Marine Microbiology, 28359 Bremen, Germany

a now at: Hessisches Landesamt für Naturschutz, Umwelt und Geologie, 65203 Wiesbaden, Germany

Correspondence: Christopher Pöhlker (c.pohlker@mpic.de) and Maria Prass (m.prass@mpic.de)





**Abstract**

The Amazon rain forest plays a major role in global hydrological cycling and biogenic aerosols are likely to influence the formation of clouds and precipitation. Information about the sources and altitude profiles of primary biological aerosol particles, however, is sparse. We used fluorescence *in situ* hybridization (FISH), a molecular biological staining technique largely unexplored in aerosol research, to investigate the sources and spatiotemporal distribution of Amazonian bioaerosols on domain level. We found wet season bioaerosol number concentrations in the range of $1 - 5 \cdot 10^5$ m$^{-3}$ accounting for >70 % of the coarse mode aerosol. Eukaryotic and bacterial particles predominated, with fractions of ~56 % and ~26 % of the intact airborne cells. Archaea occurred at very low concentrations. Vertical profiles exhibit a steep decrease of bioaerosol numbers from the understory to 325 m height on the Amazon Tall Tower Observatory, with a stronger decrease of Eukarya compared to Bacteria. Considering earlier investigations, our results can be regarded as representative for near-pristine Amazonian wet season conditions. The observed concentrations and profiles provide unprecedented insights into the sources and dispersion of different types of Amazonian bioaerosols as a solid basis for model studies on biosphere-atmosphere interactions such as bioprecipitation cycling.





## Introduction

The study of atmospheric bioaerosols represents a challenging field in aerosol research because of their diverse particle properties, including size, morphology, mixing state, hygroscopic behavior, and metabolic activity. Bioaerosols are ubiquitous in the atmosphere worldwide and comprise prokaryotic (Bacteria and Archaea) and eukaryotic (e.g., fungi and algae) cells, various reproductive entities (e.g., spores and pollen) as well as fragments of biological material (Andreae and Crutzen, 1997; Jaenicke et al., 2005; Després et al., 2012). The scientific as well as socioeconomic attention that bioaerosols have received can be explained by their manifold and fundamental roles in atmospheric chemistry and physics, biogeography, public health, ecology, and agriculture (e.g., Pöschl et al., 2010; Morris et al., 2014, Fröhlich-Nowoisky et al., 2016; Reinmuth-Selzle et al., 2017). To date, central aspects of their mechanistic roles and relevance in these fields are not fully understood or even largely unexplored. Progress in our understanding is hampered by analytical limitations in resolving the complexity, diversity, and highly dynamic life cycle of bioaerosols in the atmosphere (Morris et al., 2011; Šantl-Temkiv et al., 2019). Particularly scarce are techniques that provide atmospheric number concentrations for specific and clearly defined organism groups within the bioaerosol population.

The number of bioaerosol field observations worldwide is constantly increasing (Després et al., 2012; Fröhlich-Nowoisky et al., 2016; Šantl-Temkiv et al., 2019) with bioaerosol studies in regions that are essential for the climate system being particularly relevant. This refers to the oceans as well as forested ecosystems, which cover large areas of the Earth and entail intense surface-atmosphere interactions (e.g., Bonan, 2008; Artaxo et al., under revision; Mayol et al., 2014). Moreover, certain (though increasingly few) regions of the oceans and the large forests are still sufficiently unperturbed by man-made emissions and activities to approximate a preindustrial and, thus, pristine state of the atmosphere (Hamilton et al., 2014; Pöhlker et al., 2018). Along these lines, it has remained largely unknown which mechanistic roles "[bio]aerosols before pollution" (Andreae, 2007) have played in biogeochemical and hydrological cycles and to what extent such processes have been perturbed by the nowadays pervasive man-made emissions and activities. One important topic in this context is the ability of certain bioaerosols to act as efficient ice nuclei (IN) at comparatively warm temperatures (i.e., > -10°C) with important implications for cloud microphysics and precipitation formation (e.g., Morris et al., 2014; Delort et al., 2010).

The analytical and scientific novelty of this study is threefold: First, it widens the spectrum of techniques for bioaerosol investigations by exploring the analytical potential of fluorescence *in situ* hybridization (FISH) in this field. FISH is a molecular genetic technique for the specific staining of cells by targeting characteristic RNA or DNA sequences with complementary and fluorescently labeled nucleotide probes (e.g., Amann and Fuchs, 2008). In terrestrial and marine microbiology, FISH has become an important technique in identification and enumeration of microbial organisms with numerous applications (e.g., Pernthaler et al., 2004; Christensen et al., 1999). However, applications in bioaerosol research have remained remarkably sparse (Yoo et al., 2017; Harrison et al., 2005). Our



results demonstrate that FISH has great potential in bioaerosol analysis as it provides number concentrations of specific organism classes (i.e., from domain down to species level) and, therefore, combines bioaerosol *identification* and *quantification*. Second, this study provides number concentrations for prokaryotic and eukaryotic cells in the Amazonian rain forest atmosphere under almost pristine

conditions, which is unique data for this globally important ecosystem. In fact, the atmospheric Bacteria and Archaea concentrations are the first published results of this type for a tropical rain forest environment (Table S3). The concentrations obtained here can serve as a reference for modelling and process studies on climate-relevant forest-atmosphere interactions such as bioprecipitation-cycles. Third, this study has utilized the tall tower at the remote ATTO site to obtain vertical gradients of Bacteria,

Archaea, and Eukarya concentrations over the rain forest (with sampling heights at 5, 60, and 325 m). These gradients allow to estimate concentration ranges for bacterial, archaeal, and eukaryotic cells touching the cloud base and, thus, to assess their potential relevance for cloud microphysics.

The samples for this study were collected during prevailing clean wet season conditions in the Amazon when the bioaerosol population originates from the primary rain forest region within the

ATTO site's footprint. A detailed characterization of the sampling conditions can be found in the Supplement. The FISH protocol used in this work is an adaptation of pre-existing protocols (Glöckner et al. 1996; Pernthaler et al. 2004) with modifications and optimizations for the specific requirements of bioaerosol analysis. The main experimental steps of the FISH protocol are illustrated and (briefly) explained in Figure 1. A focal point of this study has been the careful cross-validation and comparison

of the obtained FISH results with online aerosol data as well as a synthesis with existing literature knowledge. This validation is important since FISH is experimentally demanding and prone to various artifacts (i.e. false positive or false negative counts) and thus may yield biased results (Thiele et al, 2011). Overall, we found a high consistency with complementary online data from the ATTO site as well as from previous studies, which underlines that the obtained organism concentrations are a solid

representation of the Amazonian wet season bioaerosol population.

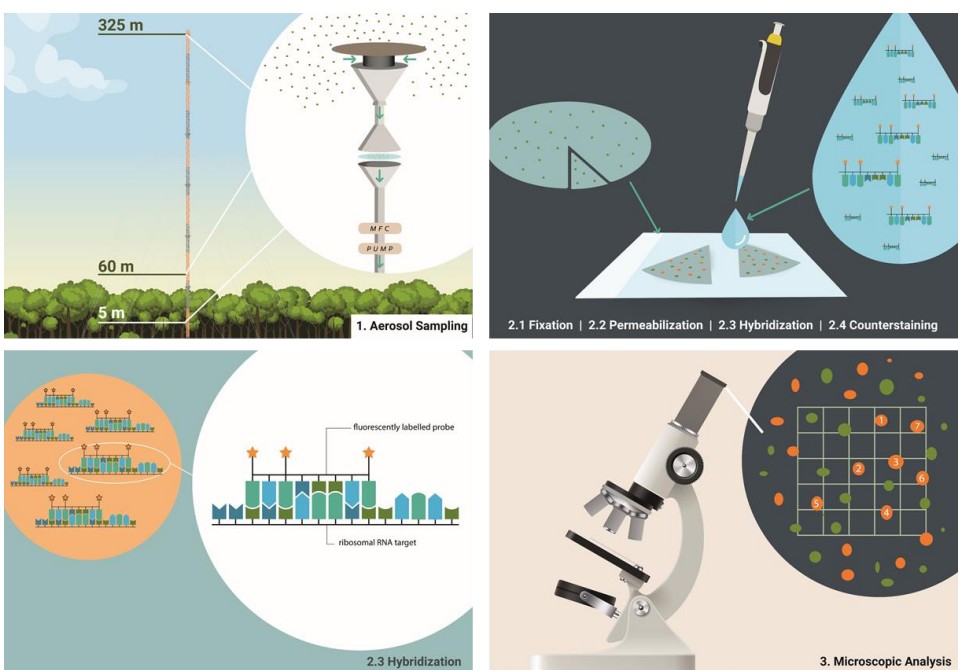

**Figure 1.** Bioaerosol sampling strategy in the Amazon rain forest and molecular genetic staining for microscopic identification and quantification. **1.** Bioaerosols were collected on polycarbonate membranes at three different sampling heights at ATTO. **2.** Biological material on the filters was prepared for staining by fixation and cell wall permeabilization. Then, fluorescently labeled oligonucleotide probes were used to assign bioaerosols at the domain level in a hybridization step. Overall bioaerosol numbers were obtained by DNA-staining with DAPI, the so called counterstaining **3.** Fluorescence signals were systematically enumerated and converted into atmospheric bioaerosol number concentrations.

**Results and discussion**

In the wet season atmosphere at the ATTO site Eukarya and Bacteria accounted for the majority of cells, whereas Archaea occurred at lower numbers and appeared to be rather rare in the investigated bioaerosols. At all sampling heights, the number concentration of eukaryotic cells ($N_{EUK}$) was highest ranging from ~3.5–38 $\cdot 10^4$ m$^{-3}$, followed by Bacteria ($N_{BAC}$) ranging from ~3.0–7.0 $\cdot 10^4$ m$^{-3}$, and Archaea ($N_{ARC}$) ranging from ~0.1–1.3 $\cdot 10^4$ m$^{-3}$ (Table 1, Figure 2). These numbers are in good agreement with estimated and measured concentrations in previous bioaerosol studies (e.g., Burrows et al., 2009b; Fröhlich-Nowoisky et al., 2016). For instance, our measured $N_{BAC}$ values fall within the estimated range of bacterial cell concentrations for forest ecosystems (i.e., 3.3–8.8 $\cdot 10^4$ m$^{-3}$) according to Burrows et al. (2009a). Of further atmospheric relevance is the number concentration of all airborne cells that were determined by staining the intracellular DNA with the fluorescent dye DAPI[1] ($N_{DAPI}$).

---

[1] DAPI = 4′,6-Diamidin-2-phenylindol is a widely used fluorescent stain for DNA.



Here, $N_{DAPI}$ ranged on average from ~12–53 $\cdot 10^4$ m$^{-3}$ (Tables 1 and 2). Due to the given specificity of the FISH probes (~80–90 % of all target cells according to the SILVAref138.1 database, www.arb-silva.de, last access 08 Dec 2020), a certain fraction of cells remains unclassified (i.e., $N_{FISH} < N_{DAPI}$ with $N_{FISH} = N_{ARC} + N_{BAC} + N_{EUK}$). In this study, $N_{FISH}$ accounted for ~60–90 % of $N_{DAPI}$ (Table 1, Supplementary Table S1), which indicates a good performance of the FISH protocol (Thiele et al., 2011, and references therein).

**Table 1.** Bioaerosol number concentrations at different heights (avg ± sd; n=5-6, samples for ~23h at each height) on domain level (Archaea, Bacteria, and Eukarya) obtained by FISH. In addition, overall bioaerosol concentrations obtained by DAPI staining. Last column shows the fraction of cells that could be assigned to one of the domains by FISH in relation to the DAPI-derived concentration.

| height | Archaea [$\cdot 10^4$ m$^{-3}$] | Bacteria [$\cdot 10^4$ m$^{-3}$] | Eukarya [$\cdot 10^4$ m$^{-3}$] | DAPI [$\cdot 10^4$ m$^{-3}$] | fraction probe/DAPI |
|---|---|---|---|---|---|
| 5 m | 0.25 ± 0.38 | 7.0 ± 2.1 | 38 ± 15 | 53 ± 21 | 0.86 |
| 60 m | 1.3 ± 1.2 | 6.5 ± 2.5 | 14 ± 3.3 | 25 ± 10 | 0.85 |
| 325 m | 0.10 ± 0.21 | 3.0 ± 1.3 | 3.5 ± 1.2 | 12 ± 4.6 | 0.61 |

Figure 2 shows the time series of $N_{EUK}$, $N_{BAC}$, $N_{ARC}$, and $N_{DAPI}$ at 60 m height with complementary meteorological and aerosol data under pristine rain forest conditions. Here, the total aerosol particle count between ~0.7 and 10 µm ($N_{0.7-10}$) – corresponding to the effectively DAPI- and FISH-counted size range – serves as a reference number concentration and ranges from ~30–48 $\cdot 10^4$ m$^{-3}$ (Table 2). Relative to $N_{DAPI}$, Eukaryotes accounted on average for ~56 %, Bacteria for ~26 %, and Archaea for ~5 % of the cells. The bioaerosol number concentrations $N_{EUK}$, $N_{BAC}$, $N_{ARC}$, and $N_{DAPI}$ show a clear day-to-day variability: For instance, $N_{EUK}$ varies by a factor of 2, whereas $N_{BAC}$ varies by a factor of 4 (Table S1). $N_{ARC}$ shows even larger variations, although the low counting statistics here require caution in interpreting these results.[2] Along these lines, also the bioaerosol mixture – i.e., the ratios of $N_{EUK}$, $N_{BAC}$, and $N_{ARC}$ relative to $N_{DAPI}$ as represented by the Pie charts in Figure 2 – show a clear variability. Here the days from 1 to 3 Mar 2018 stand out as they are characterized by a rather high abundance of $N_{BAC}$. This increase in $N_{BAC}$, might be related to the strong rain event in the night from 27 to 28 Feb 2018. Bacterial cells on the leaf surfaces might have been emitted through mechanical momentum of the raindrop impaction according to Joung et al. (2017) and/or might be related to a "post-rain" bioaerosol enhancement according to Huffman et al. (2013). While the initial results presented here emphasize such potential links between the variability in bioaerosol concentrations and meteorological environmental parameters (which are speculative so far), the statistical basis of these initial FISH results is too small to constrain these relationships. Accordingly, an investigation of bioaerosol

---

[2] In fact, we refrain from interpreting $N_{ARC}$ in great detail in this work due to the low statistics. Furthermore, the probe ARCH915 used here was found to hybridize with some Bacteria, which could lead to false-positive signals.





emission mechanisms in relation to the local and regional meteorology requires more extended follow-up FISH studies.

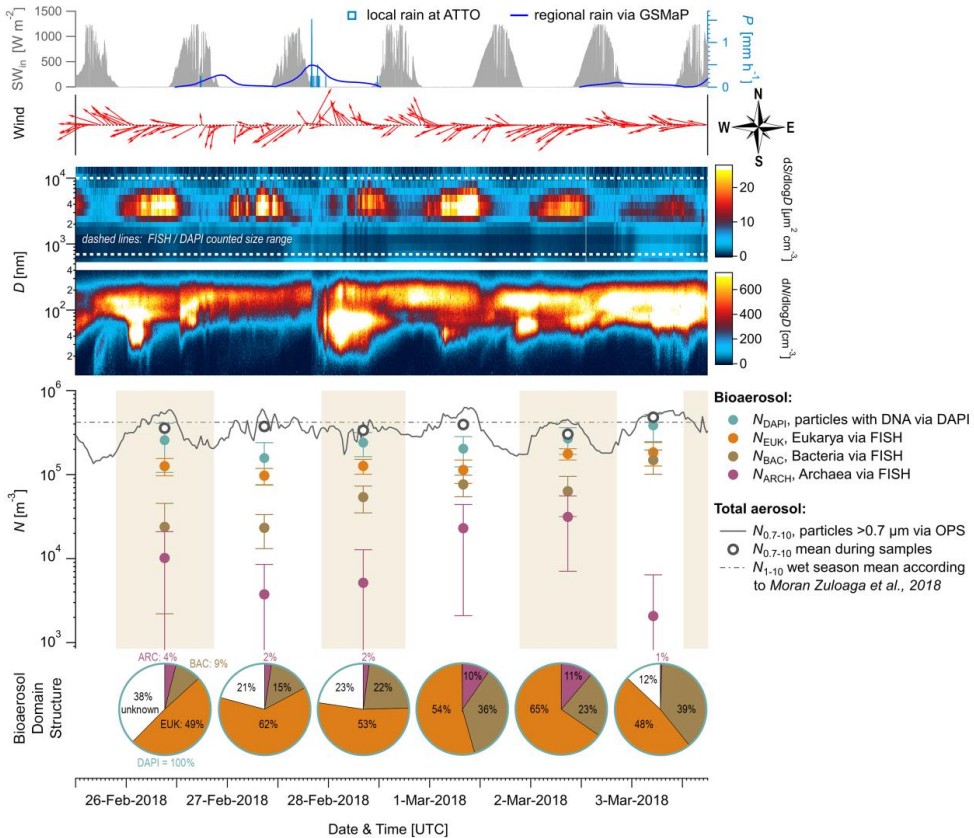

**Figure 2.** Time series of aerosol number concentrations and complementary aerosol and meteorological data at 60 m height, observed over six days during the wet season 2018. From top to bottom: i) meteorological data including incoming solar radiation (SW$_{in}$, grey shaded), precipitation rates (P, blue curve and bars), and wind vectors (red arrows) ii) contour plots displaying total aerosol number size distributions obtained by a Scanning Mobility Particle Sizer (0.01 to 0.4 µm) and an Optical Particle Sizer (0.5 to 10 µm) iii) bioaerosol number concentrations at the domain level from FISH and DAPI staining (markers as mean and error bars as one standard deviation) with shaded areas as filter sampling periods (each approx. 23 h), and iv) pie charts showing daily bioaerosol mixture based on number concentrations at the domain level.

In addition to intact airborne cells, bioaerosol definitions also include biological fragments (Després et al., 2012). These fragments – a complex mixture of biological material in a continuum of degradation states, e.g., from mechanical fragmentation, cell rupture, or cytosol release – can be of



significant atmospheric relevance as they may comprise (high) ice activity or allergenic potential (Šantl-Temkiv et al., 2015; Steiner et al., 2015; Reinmuth-Selzle et al., 2017). However, a direct analysis of these fragments is often notoriously difficult because of their morphologically and biologically undefined state. Both, the DAPI and FISH quantifications predominantly target intact cells, since

upon cell rupture or damage the contained nucleic acids might be released and degraded. Therefore, the ratio of $N_{DAPI}$ vs. $N_{0.7-10}$ provides a valuable estimate of the presumably intact cell fraction vs. the fraction of fragments within the size range from 0.7 to 10 µm of the Amazonian bioaerosol population. This estimate relies on the assumption that under unperturbed wet season conditions the vast majority of coarse mode particles originates more or less directly from primary emissions of the rain for-

est (compare Moran-Zuloaga et al., 2018; Pöhlker et al., 2018). This assumption is justified here since other potential coarse mode sources (i.e., Saharan dust, Atlantic sea salt, and ash from biomass burning) can be largely excluded during the sampling period. On average, intact cells accounted for the majority of coarse mode particles with $N_{DAPI}/N_{0.7-10}$ values of ~70 %, being in good agreement with previous studies (Table 2 and S3). Accordingly, we obtained ~30 % on average as an upper limit esti-

mate for the fraction of fragments and degraded biological material in this size range. The estimated concentration ($3–19 \cdot 10^4 \, m^{-3}$) and fraction (12–58 %) of fragments is quite variable, which points at interesting open questions for follow-up studies on potential degradation pathways in the Amazonian bioaerosol cycling.

**Table 2.** Mean diel aerosol number concentrations at 60 m height obtained by an optical particle sizer (OPS) and by bioaerosol staining with DAPI (avg ± sd). The fraction of DAPI-stained particles in relation to total aerosol numbers in the same size range provides an estimation of presumably intact cells versus degraded biological material.

| sample | OPS [$\cdot 10^{-4} \, m^{-3}$] | DAPI [$\cdot 10^{-4} \, m^{-3}$] | fraction DAPI/OPS |
|---|---|---|---|
| day 1 | 36 ± 13 | 26 ± 15 | 0.72 |
| day 2 | 37 ± 9.7 | 16 ± 8.2 | 0.42 |
| day 3 | 34 ± 5.9 | 24 ± 7.7 | 0.71 |
| day 4 | 39 ± 14 | 20 ± 8.0 | 0.52 |
| day 5 | 30 ± 10 | 27 ± 9.4 | 0.88 |
| day 6 | 48 ± 9.1 | 38 ± 14 | 0.80 |
| avg (1-6) | 37 ± 10 | 25 ± 10 | 0.67 |




Furthermore, we investigated the bioaerosol variability with height across the lower 325 m of the boundary layer to assess the gradients of specific organism classes in this particularly important part of the atmospheric vertical structure. As expected, the vertical profiles displayed in Figure 3 show a general and rather steep decrease in the average cell concentrations ranging from $N_{\mathrm{DAPI}} = 53 \cdot 10^4$ m$^{-3}$

at 5 m, via $25 \cdot 10^4$ m$^{-3}$ at 60 m (a 52 % reduction) to $12 \cdot 10^4$ m$^{-3}$ at 325 m (77 % reduction compared to 5 m). The eukaryotic cell concentration, $N_{\mathrm{EUK}}$, shows a similarly steep decrease in its profile. For bacterial cells, however, we found a less steep vertical trend with similar concentrations at 5 m and 60 m ($\sim 7.1 \cdot 10^4$ m$^{-3}$ vs. $6.5 \cdot 10^4$ m$^{-3}$), followed by a 54 % reduction from 60 m to 325 m ($\sim 3 \cdot 10^4$ m$^{-3}$). For Archaea, the highest concentrations were found at 60 m, although the low concentrations and

counting statistics do not allow robust conclusions on the vertical profile of this organism class. All concentrations are summarized in Table 1. As an additional aspect, we also calculated airborne DNA mass concentrations based on aforementioned FISH number concentrations in combination with typical mean genome sizes of fungi, Bacteria and Archaea.[3] Such results on atmospheric DNA concentrations are sparse and typically based on photometric DNA quantification after extraction from aerosol

filters. We obtained average airborne DNA mass concentrations of 11.9 ng m$^{-3}$ at 5 m, 4.5 ng m$^{-3}$ at 60 m, and 1.2 ng m$^{-3}$ at 325 m (Table S2). In general, these results are comparable to studies conducted at an urban site yielding 7 ng m$^{-3}$ (Després et al., 2007), a boreal forest yielding $8.60 \pm 11.1$ ng m$^{-3}$ (Helin et al., 2017), and the tropical region of Singapore yielding 0.69 to 6.9 ng m$^{-3}$ (Gusareva et al., 2019). The Amazonian DNA concentrations presented here can be considered as a

lower limit (for details, see appendix). Our data suggests that the Amazonian air microbiome to host larger quantities of DNA mass concentration, than reported for other ecosystems before.

The clear difference in the $N_{\mathrm{EUK}}$ vs. $N_{\mathrm{BAC}}$ profile structures might be due to different distribution of the organism sources inside and below the canopy space (i.e., biofilms on leaves according to Morris et al., 1997 vs. pronounced fungal spore emission at the ground according to Elbert et al., 2007 and

Löbs et al., 2020). Another reason might be the different sedimentation velocities and, thus, airborne residence times, due to widely different particle mass. Further, please note that the fraction of unclassified particles increased substantially towards 325 m, which may be related to enhanced cell aging due to radiation and/or atmospheric oxidation upon upward transport. Typical cloud base heights in the central Amazon range between 500 and 1500 m (O. Lauer, personal communication, 2020), which

is substantially higher than the 325 m sampling height used here. Still, the measured values for $N_{\mathrm{EUK}}$, $N_{\mathrm{BAC}}$, $N_{\mathrm{ARC}}$, and $N_{\mathrm{DAPI}}$ at 325 m can serve as a solid upper limit estimate for cell concentrations being convectively lifted to cloud base. This estimate may be of value for cloud microphysical process studies in combination with Amazonian IN observations and parameterizations (e.g., Prenni et al., 2009; Schrod et al., 2020).

---

[3] With the chosen approach, this quantification exclusively accounts for intracellular DNA and omits the fraction of extracellular DNA.

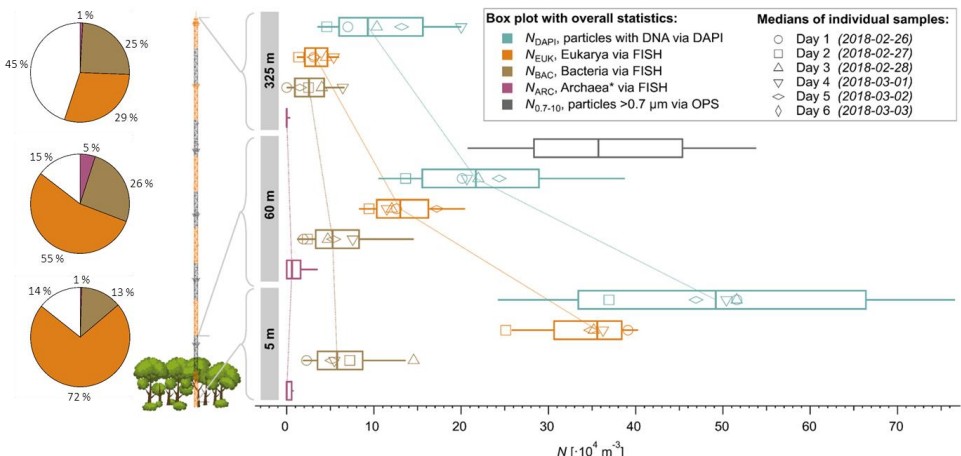

**Figure 3.** Height profiles of aerosol number concentrations observed at 5 m, 60 m, and 325 m above ground at the ATTO-tower. The median concentrations of all daily samples are displayed as vertical box-whisker plots with 25 and 75 quartiles as box, and 10 and 90 percentiles as whiskers. Daily median values are shown as markers according to the legend. Bioaerosols were quantified with FISH as well as DAPI staining. The total aerosol number concentration at 60 m was determined by an optical particle sizer (OPS) in the corresponding size range.

Finally, the microscopic enumeration of cells after staining also provides qualitative insights into the mixing state of the Amazonian bioaerosol population, which is an important aspect of the Amazonian bioaerosol cycling (Pöschl et al., 2010). Figure 4 shows typical fluorescence images after DAPI and FISH staining obtained from the three different sampling heights. Figure 4G and H show an example of a cell agglomerate comprising multiple eukaryotic and bacterial cells. The vast majority of cells, however, was observed as separated cells, which suggests that under the given wet season conditions the bioaerosol components are largely externally mixed. Another (qualitative) observation worth noting is a decrease of average cell size with height (i.e., more larger particles with > 2 µm at 5 and 60 m relative to 325 m). Several of the large cells could be identified as fungal and fern spores, based on morphological criteria. An enrichment of larger particles at the lower heights is in accordance with their higher sedimentation tendency and lower atmospheric residence times. The decreasing size with height corresponds well to the increasing fraction of bacterial cells, which are typically smaller than eukaryotic cells. A systematic retrieval of bioaerosol number size distributions from the FISH micrographs to investigate such trends in details, however, is rather challenging and thus will be subject of a follow-up study.

**Biogeosciences** Open Access
Discussions
EGU

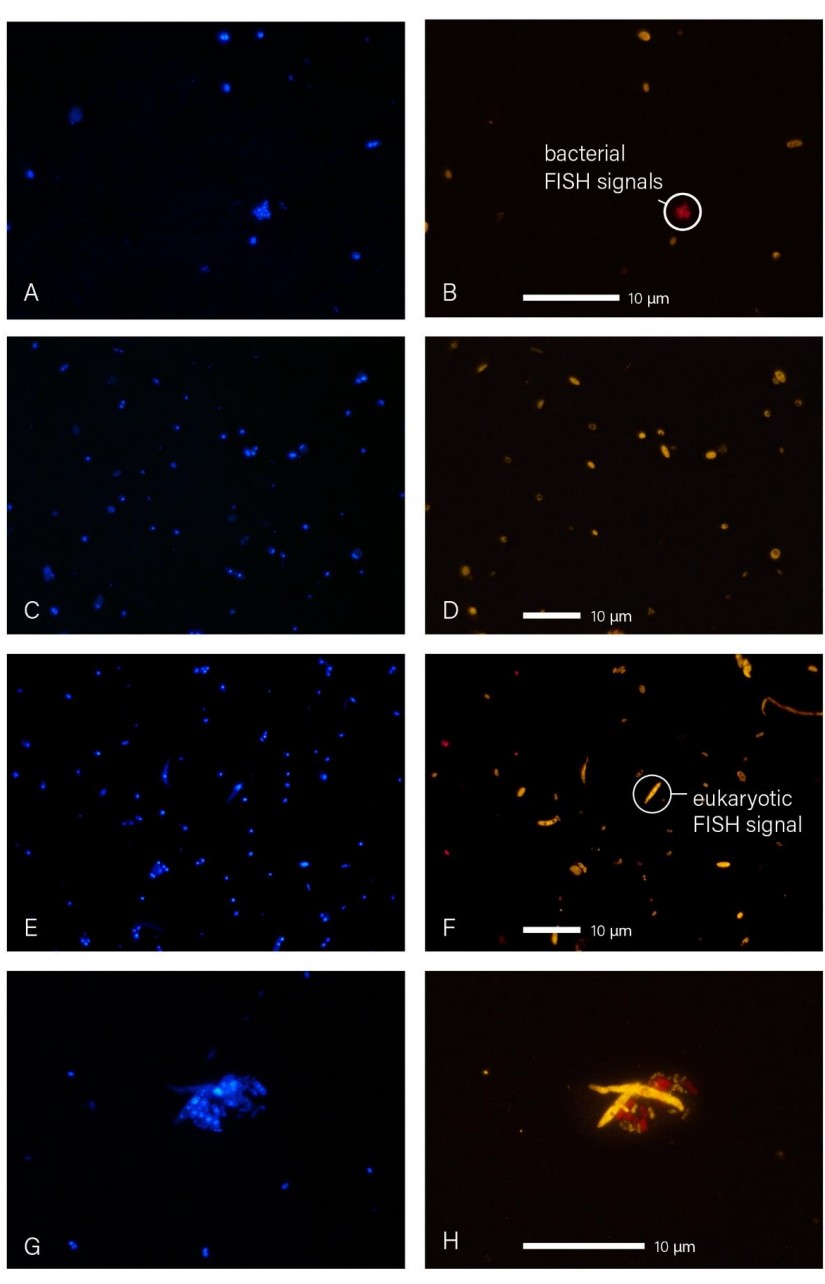

**Figure 4.** Microscopic images of fluorescence signals after DNA staining with DAPI (left panel, blue) and FISH (right panel, eukaryotes in orange and bacteria in red). Filter samples displayed here were collected at 325 m (A, B), 60 m (C, D), and 5 m (E, F, G, H). Particle agglomerates, as shown in G and H, we found rather rarely. Agglomerate here shows cluster of fungal spores and bacterial cells.



**Conclusions**

Our study showed that FISH has great analytical potential in bioaerosol analysis. It combines bioaerosol identification and quantification and, thus, provides insights into the concentration levels and spatiotemporal variability of specific and clearly defined organism groups within the bioaerosol population. We investigated the Amazonian bioaerosols on domain level by quantifying eukaryotic, bacterial, and archaeal cells as well as the overall concentrations of airborne cells as a function of time and height within and above the forest canopy. The results characterize bioaerosols during clean wet-season conditions in the Amazon under the predominance of local emissions from the primary rain forest. Eukarya (29–72 %) and Bacteria (13–26 %) dominated the bioaerosol population and variability, whereas Archaea (0,5- 5 %) played numerically only a minor role. On average ~70 % of the coarse mode particles (i.e., 0.7–10 µm) were attributed to presumably intact cells whereas the remaining ~30 % can be regarded as an upper limit estimate for biological fragments and degenerated biological material in this size fraction under the given conditions. The bioaerosol concentrations decreased substantially with height with ~2 times less Bacteria and even ~10 times less Eukaryotes at 325 m than under the canopy. This emphasizes the importance of the sampling height as a variable for bioaerosol observations in the Amazon. The different shapes of the bacterial vs. eukaryotic concentration profiles may be attributed to different source locations in and below the canopy and/or differences in aerodynamic mobility of the cells upon vertical transport. Overall, the results of this study greatly extend the knowledge on the life cycle of the Amazonian aerosols and provide a solid experimental basis for model investigations of bioaerosol-related processes, such as the role of biological ice nuclei or giant cloud condensation nuclei in cloud microphysics and potential bio-precipitation cycling.

Pronounced diurnal patterns with a maximum of coarse mode particle abundance during the night represent a characteristic feature of Amazonian aerosol cycling (Figure 2). The current study demonstrates that a dedicated FISH analysis with separated day vs. night sampling promises to resolve and quantify the organism classes that constitute the diurnal pattern. This further relates to the open question on the main meteorological drivers for bioaerosol emissions in the rain forest ecosystem. For this purpose, a broader statistical basis of FISH results along with meteorological observations is needed. Essential for microphysical bioaerosol analyses is a retrieval of the number size distributions from the DAPI and FISH data sets. Fluxes of specific organism classes from the forest could potentially be determined with a dedicated FISH sampling during periods of strong convection. In addition, the wet season characterization presented here requires a complementary dry season sampling to resolve potential seasonal differences in the bioaerosol abundance and mixture. Finally, the taxonomic resolution of this study operates exclusively on domain level. Future studies should use the analytical potential of FISH by targeting organism classes on lower taxonomic levels (e.g., theoretically down to species level). This is of particular interest in terms of differences in IN activity influencing the formation of clouds. In a bigger picture, we envision that dedicated FISH studies may be conducted in close relation to cloud microphysical process studies. Targeted bioaerosol characterizations during periods of





climate extremes, such as El Niño-related droughts in the Amazon, would be of great importance to study the response and resilience of the bioaerosol population in the Amazon under warmer and presumably drier climatic conditions in the future.



## Appendix: Materials and Methods

### *Measurement location: The Amazon Tall Tower Observatory:*

The Amazon Tall Tower Observatory is a research site located in the Uatumã Sustainable Develop-
ment Reserve, Amazonas State, Brazil (Andreae et al., 2015). It comprises several ground-based con-
tainers and three towers of different heights (80 m height: 'Triangular mast' and 'Walk-up tower'; 325
m height: 'Tall tower')"equipped with state-of-the-art instrumentation to analyze biosphere-atmos-
phere exchange processes in this remote continental location. The forest ecosystem is driven by alter-
nating wet and dry seasons inducing conditions that temporarily resemble a pre-industrial and thus
pristine state. Hundreds of square kilometers of untouched primary forest surround the research sta-
tion, forming its biogeochemical footprint region (Pöhlker et al., 2019). Further information on the
sampling location can be found in the supplement.

### *Aerosol sampling at ATTO*

This study focuses on seven aerosol samples, collected during the wet season from 25 Feb 2018 to 3
Mar 2018, with an approximate sampling duration of 23 h each. Samples at 5 m and 60 m height were
collected at the triangular mast, those at 325 m at the tall tower. At 5 m, the filter holder was con-
nected directly to a total suspended particle (TSP) inlet. At 60 m and 325 m height, filter holders were
mounted in a ground-based container and connected to a TSP inlet via stainless steel inlet lines. Aero-
sols were filtered onto white polycarbonate membranes (Isopore PC Membrane, 0.2 µm pore size, 47
mm diameter, GTTP04700, Merck, Darmstadt, Germany) by applying a vacuum. Filters were auto-
claved at 121°C and 220 kPa for 15 min before use. The sample air flow rate was set to 9 lpm by a
digital mass flow controller (D-6341-FGD-22-AV-99-D-S-DR, Wagner Mess- und Regeltechnik, Of-
fenbach am Main, Germany) installed between the pump (N840.3FT.18, KNF Neuberger, Freiburg im
Breisgau, Germany) and a custom-made filter holder.

### *Complementary online measurements at ATTO*

During filter sampling, three instruments measured aerosol number concentrations in parallel at 60 m
height: an Optical Particle Sizer (OPS, model 3330, size range 0.3–10 µm, TSI Inc., Shoreview, MN,
USA) and a Scanning Mobility Particle Sizer (SMPS, classifier 3080, detector 3722, DMA 3081, size
range: 0.01–0.42 µm, TSI Inc., Shoreview, USA), both detected aerosols in a size resolved manner,
and a Condensation Particle Counter (CPC, model 5412, Grimm Aerosol, Ainring, Germany) meas-
ured total aerosol concentrations. Detailed information on OPS, SMPS, and CPC measurements can
be found in Andreae et al. (2015) and Moran-Zuloaga et al. (2018). Stained bioaerosols could only be
detected and identified as such by microscopy if their diameter was ~0.7µm or larger. For a compari-
son between $N_{FISH}$ and total aerosol numbers, only OPS data detected in the according channels was
considered (0.74–10 µm, $N_{0.7-10}$). Several sensors monitored meteorological conditions at ATTO such
as incoming shortwave radiation (Pyranometer, CMP21, Kipp & Zonen, Netherlands) and rainfall



(Rain gauge, TB4, Hydrological Services Pty. Ltd., Australia). Further information on micrometeorological sensors and instrumentation at ATTO can be found in Andreae et al. (2015).

### *Fluorescence in Situ Hybridization*

Several previous studies containing fluorescence *in situ* hybridization (FISH) protocols were considered in terms of buffer ingredients, incubation times, and further details, to obtain reliable results in bioaerosol analysis. Original references can be found in Glöckner et al. (1996, 1999), Pernthaler et al. (2004), Fuchs et al. (2007), and Schmidt et al. (2012). The chemicals used for fixation, permeabilization, hybridization, staining, and mounting are listed in the supplement (Table S4). Best results were

obtained by applying the following procedure: Directly after sampling, bioaerosols on the filters were fixed by an incubation in a freshly prepared solution of 2 % Formaldehyde in phosphate buffered saline (PBS). For this purpose, filters were inserted into glass filtration towers (107003970, Sartorius, Göttingen, Germany) and covered with ~15 mL of the solution. The liquid was removed after 1 h at 28°C ambient temperature by applying a gentle vacuum. Subsequently, filters were flushed by cover-

ing them with 20 mL deionized water (MQ water) and applying vacuum again. The same procedure was repeated with 20 mL Ethanol 70 %. Filters were air-dried and stored in Analyslide petri dishes (7231, Pall corporation, New York, USA) at - 20°C in the freezer. Filters were transported to Germany frozen and stored in the freezer at -20°C until further processing.

The filters were then cut into sections and numbered with a pencil at room temperature. For each

sample, one fixed, cut and numbered filter section was directly mounted in Citifluor AF1 (Citifluor Ltd., Canterbury, UK) containing 4 µg mL$^{-1}$ DAPI (4′,6-Diamidin-2-phenylindol, Serva, Heidelberg, Germany) for total cell number detection.

To prevent cell loss during FISH, filter sections were covered with a thin layer of low gelling point agarose (0.2 % in MQ water). Cell wall permeabilization by means of incubation in lysozyme solution

(10 mg mL$^{-1}$; 60 min for EUK516 and 45 min for EUB338-I-III (EUB-mix), ARCH915, and NON338) and achromopeptidase solution (60 U ml$^{-1}$, 20 min for EUB338-mix, ARCH915, and 338), both at 37°C, enabled the entrance of oligonucleotide probes during hybridization. To remove all enzymes, filter sections were washed in excess MQ water. Subsequently, the filter sections were incubated in 30 µL hybridization buffer (900 mM NaCl, 20 mM Tris/HCl, 1 % blocking reagent, 0,01 %

SDS, and formamide depending on probe) containing 2 µL probe working solution (8.4 pmol µl$^{-1}$) at 46°C for 120 min. Oligonucleotide probes targeting bacterial, eukaryotic, and archaeal cells were used. The probe NON338 served as negative control. Probe sequences, labels, and the respective formamide concentrations are presented in Table A1. After hybridization, filter sections were directly transferred into 50 mL preheated washing buffer (0.9 M (EUK516) or 0.08 M (EUB338-mix,

ARCH915, NON338) NaCl, 20 mM Tris/HCl (pH 7.4), 5 mM EDTA, and 0.01 % SDS) and incubated freely floating for 15 min at 48°C in the dark. The 50 mL tubes containing washing buffer and





filter sections were gently inverted when the incubation started and ended. Subsequently, filter sections were rinsed in a Petri dish containing MQ water and a second Petri dish containing 70 % ethanol. Filter sections were dried on Kim wipes at room temperature for 15-30 min. Dry filter sections were mounted in Citifluor AF1 containing 4 µg mL$^{-1}$ DAPI.

*Epifluorescent microscopic enumeration and bioaerosol projection*

Filter sections were inspected with a Nikon Ti2-E inverse epifluorescence microscope (Nikon, Microscope Solutions, Minato, Japan) at 600x magnification (objective: Apo Lamda S 60x Oil with 1.4 numerical aperture and a 10x widefield ocular). Epifluorescence filter cubes were chosen according to

the fluorescent dye properties as summarized in Table A1. Fluorescence signals deriving from DAPI staining or FISH were counted manually using an ocular grid (Zeder et al., 2011). One person examined all filter samples for FISH and DAPI signals to ensure consistent counting procedure. As in previous studies, the examiner rested regularly to avoid eye fatigue leading to decreasing signal detection. FISH and DAPI signals were detected as such, by taking their color, fluorescence intensity, size,

shape, and surface structure into account. Raw counts were documented with help of a mechanical counter. In a first step, filter sections that were embedded in a mix of Citifluor and DAPI were analyzed. The atmospheric number concentrations of bioaerosols that were stained with the DNA-dye were calculated by extrapolating DAPI raw counts with respect to the grid size, covered filter area and sampled air volume.

$$N_{\mathrm{DAPI}} = \frac{\mathrm{N_{grid} \cdot A_f}}{\mathrm{A_{grid}} \cdot V_{\mathrm{air}}}$$

$N_{\mathrm{DAPI}}$ = atmospheric bioaerosol number concentration stained with DAPI [m$^{-3}$]

$\mathrm{N_{grid}}$ = number of DAPI stained cells counted per grid

$\mathrm{A_f}$ = area filter [mm$^2$]

$\mathrm{A_{grid}}$ = area grid [mm$^2$]

$\mathrm{V_{air}}$ = sampled air volume [m$^3$]

Afterwards, filter sections treated with the FISH technique were inspected. The FISH signals were enumerated first ($N_{\mathrm{FISH}}$), and consecutively DAPI counterstaining signals were quantified in the same

field of view to avoid bleaching of the former. Ratios of hybridized bioaerosols were calculated and multiplied with the bioaerosol number concentrations obtained by DAPI staining only. To achieve robust statistics at least 500 DAPI stained cells per filter sample and probe were inspected, often more than 1000 were counted. According to Pernthaler et al. (2003) this reduces the counting error to <5 %. Raw counts of hybridized and DAPI stained cells for each filter sample are presented in the supple-

ment (Table S1).





*Quantification of atmospheric DNA concentration*

Airborne DNA mass was calculated by multiplication of mean bioaerosol numbers obtained by FISH with the typical DNA mass of a bacterial, eukaryotic, or archaeal cell.

$$m_{\text{DNA}} = \frac{N_{FISH} \cdot \text{bp} \cdot 609.7 \ \text{g/mol}}{N_A}$$

$m_{\text{DNA}}$ = airborne DNA mass [g m$^{-3}$]

$N_{\text{FISH}}$ = bioaerosol number concentration obtained by FISH [1 m$^{-3}$]

bp = genome size [base pair cell$^{-1}$]

10      609.7 g mol$^{-1}$ = average mass of a base pair in bound form (see appendix)

$N_{\text{A}}$ = Avogadro constant

The genome size of Bacteria was defined as 4 Mb, as found in the NCBI data base for airborne Bacteria present in the Amazonian air microbiome (e.g. Proteobacteria, Actinobacteria, Souza et al.,

15      2019). Archaeal genome size was defined as 4 Mb as well (Landenmark et al., 2015). The genome size of fungi was used as a representative value for Eukaryotes, since coarse mode bioaerosols in the Amazon were reported to mainly consist of fungal spores (Graham et al., 2003; Huffman et al., 2012). As these genome sizes are several orders of magnitude smaller compared to those of higher plants, we consider the here presented airborne DNA mass obtained this way as a lower limit for the Amazon

20      forest bioaerosol. In NCBI the typical genome size for basidiomycetes and ascomycetes is indicated as 30 Mb.





**Table A1:** Technical details of rRNA targeting probes and corresponding microscopic filters (excitation, dicroic mirror, and emission) used for FISH. As described in Daims et al. (1999), a mixture of EUB338 I, II, and III (referred to as EUB-mix) was applied for identification of Bacteria. By use of ARCH915, Archaea were identified and EUK516 was applied to hybridize Eukarya. NON338 served as negative control. DAPI stains all particles containing DNA by attaching preferably to adenine and thymine rich sequences. For our experiments, fluorescent labels in the reddish wavelength range were chosen to avoid overlap with the autofluorescence of bioaerosols which is typically strong in the green wavelength range (Pöhlker et al., 2012).

| probe/stain | sequence/ target | label | form-amide | reference | exc. | dic. mirror | em. |
|---|---|---|---|---|---|---|---|
| EUB338I | GCTGCCTCCCGTAGGAGT | 4x ATTO594 | 35% | Amann et al. 1990 | | | |
| EUB338II | GCAGCCACCCGTAGGTGT | 4x ATTO594 | 35% | Daims et al. 1999 | | | |
| EUB338III | GCTGCCACCCGTAGGTGT | 4x ATTO594 | 35% | Daims et al. 1999 | 562/40 | 593 | 624/40 |
| NON338 | ACTCCTACGGGAGGCAGC | 4x ATTO594 | 35% | Wallner et al. 1993 | | | |
| ARCH915 | GTGCTCCCCCGCCAATTCCT | 1x ATTO594 | 35% | Stahl and Amann, 1991 | | | |
| EUK516 | ACCAGACTTGCCCTCC | 1x ATTO542 | 0% | Amann et al. 1990 | 545/25 | 565 | 605/70 |
| DAPI | DNA | | | | 387/11 | 400 | 409 LP |



**Data availability.** Online ATTO data can be found in the ATTO data portal under https://www.atto-data.org/ (ATTO, 2020). All essential results from FISH and DAPI staining are provided in the main text and supplementary tables. For data requests beyond the available data, please refer to the corresponding authors.

**Author contributions.** MP and CP developed the research and the experiments. MP conducted the sampling, the sample analysis and manuscript preparation with contributions from all co-authors. FD, LAK, MAF, ACA supported the sample collection, data acquisition at the ATTO site, and online data analysis. IHA and JW supported in the laboratory filter analysis. BMF provided advice and guidance for the adaptation of FISH protocols for bioaerosol analysis. DW and TK developed data analysis rou-
tines. MOA, JK, BW, PA, and ET provided valuable ideas to the data analysis and interpretation. WE contributed to the literature research and comparison with earlier studies. CP and UP supervised the work. All the authors contributed to the interpretation of the results and writing of the paper.

**Competing interests.** The authors declare that they have no conflict of interest.

**Acknowledgements.** This work has been supported by the Max Planck Society (MPG) and the Max
Planck Graduate Center with the Johannes Gutenberg University Mainz (MPGC). For the operation of the ATTO site, we acknowledge the support by the German Federal Ministry of Education and Research (BMBF contract nos. 01LB1001A and 01LK1602B) and the Brazilian Ministério da Ciência, Tecnologia e Inovação (MCTI/FINEP contract 01.11.01248.00) as well as the Amazon State University (UEA), FAPEAM, LBA/INPA, and SDS/CEUC/RDS-Uatumã. This paper contains results of re-
search conducted under the Technical/Scientific Cooperation Agreement between the National Institute for Amazonian Research, the State University of Amazonas, and the Max-Planck-Gesellschaft e.V.; the opinions expressed are the entire responsibility of the authors and not of the participating institutions. We acknowledge the support by the Instituto Nacional de Pesquisas da Amazônia (INPA). We would like to thank Reiner Ditz, Stefan Wolff, Susan Trumbore, Alberto Quesada, Hermes Braga
Xavier, Andrew Crozier, Nagib Alberto de Castro Souza, Thiago de Lima Xavier, Thomas Disper, Josué Ferreira de Souza, Feliciano de Souza Coelho, Antonio Huxley Melo Nascimento, André Luiz Matos, Elton Mendes da Silva, Björn Nillius, Antonio Ocimar Manzi, Roberta Pereira de Souza, Wallace Rabelo Costa Amauri Rodriguês Perreira, Steffen Schmidt, Uwe Schulz, Bruno Takeshi, and Adir Vasconcelos Brandão for technical, logistical, and scientific support within the ATTO project.
We thank Annemarie Zahn for graphical support with figure design. We thank especially Jörg Wulf, Andreas Ellrott, Stefan Thiele, and Rudolf Amann for substantial scientific and technical support with the FISH analysis. Moreover, we thank Jan-David Förster, Viviane Després, Janine Fröhlich-Nowoisky, Thomas Behrendt, Anna Kunert, Ovid Krüger, Oliver Lauer, Bruna A. Holanda, Matthias Sörgel, and Luiz A. T. Machado for scientific support and stimulating discussions.






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
