# Peer review of "Bioaerosols in the Amazon rain forest: Temporal variations and vertical profiles of Eukarya, Bacteria and Archaea"

_Biogeosciences, 2020_

## Author Comment (AC1)

**Manuscript format description:**

Black text shows the original referee comment, blue text shows the authors response, and red text shows quoted manuscript text. Changes to the manuscript text are shown as *italicized and underlined*. We used bracketed comment numbers for referee comments (e.g., [R1.1]) and author's responses (e.g., [A1.1]). Line numbers refer to the discussion/review manuscript.

**Anonymous Referee #1,**

Received: 12 Feb 2021

Prass et al. provide an analysis of the domain level diversity of bioaerosols from the Amazon rainforest. This dataset provides a window into an area of the atmosphere considered to be less affected by human pollution, which could help deconvolve what microbes are "naturally" in the atmosphere vs there by human-activities. It also provides a dataset that focuses on enact cells compared to sequencing methods that could also include eDNA.

We appreciate the constructive comments by Referee #1, which have been considered carefully and helped to improve the quality of our manuscript. The referee's comments and our responses are outlined in detail below:

**Major comments:**

[R1.1] The manuscript would be improved by more discussion of other bioaerosol-microbe focused papers and how they relate to the current findings, such as Souza et al. 2019 (referenced but not discussed), Stern et al. 2021; Env. Sci.&Tech. Zhen, Sci. Total Environ. 2017; Yamaguchi, N. Sci. Rep. 2012. What is known about the different domains and their potential ecosystem roles or residence times?

[A1.1] Thanks for the suggestion. More papers on the role of fungal and bacterial bioaerosols were implemented in the discussion as specified in detail below:

(P5, L16-P6, L3): These numbers are in good agreement with estimated and measured concentrations in previous bioaerosol studies (e.g., Burrows et al., 2009b; Fröhlich-Nowoisky et al., 2016). For instance, our measured $N_{BAC}$ values fall within the estimated range of bacterial cell concentrations for forest ecosystems (i.e., $3.3–8.8 \cdot 10^4$ m$^{-3}$) according to Burrows et al. (2009a). *Furthermore, a predominance of Eukaryotes in the Amazon was shown before e.g. by Souza et al. (2019) and Elbert et al. (2007), which is consistent with our results.*

(P6, L28-P7, L13): While the results presented here emphasize such potential links between the variability in bioaerosol concentrations and meteorological environmental parameters (which are speculative so far), the statistical basis of these initial FISH results is too small to constrain these relationships. Accordingly, an investigation of bioaerosol emission mechanisms in relation to the local and regional meteorology requires more extended follow-up FISH studies. *In contrast to the bioaerosol burden mainly originating from forest emission during clean wet season conditions, an investigation of long-range transport-related changes in the air microbiome might be of interest, for instance with respect to dust-associated bacteria as found by Yamaguchi et al. (2018) and Prospero et al. (2005).*

[R1.2] The study seems to pause right when something unique to FISH (the future work proposed at the end of page 10) could be presented. It is unclear why only FISH was utilized. The current standard for airborne microbiology appears to be sequencing based (see papers in point 1), which provides higher taxonomic resolution than the domain level FISH analysis conducted in this manuscript. 16S rRNA qPCR is more robust than microscopy for counting as well. The manuscript lacks an explanation why sequencing techniques were not applicable for this system (why wasn't FISH and sequencing done?) and should justify the decision to only look at domain level diversity (why weren't more FISH probes used?).

[A1.2] We thank Reviewer #1 for this comment. The reviewer questions here several aspects that are obviously not clear enough in the current version of the manuscript. Accordingly, we took this criticism into account to further clarify and sharpen our argumentation. Specifically, we added a new section into the supplement (S1.4), which address and justify in detail the various aspects brought up by the reviewers. We further adjusted and optimized several statements in the main text of the manuscript. In [R.1.2], the reviewer combines several questions, which are answered separately below.

"It is unclear why only FISH was utilized. The current standard for airborne microbiology appears to be sequencing based (see papers in point 1), which provides higher taxonomic resolution than the domain level FISH analysis conducted in this manuscript.

We are aware that sequencing-based techniques provide a higher taxonomic resolution. The aim of this study, however, was not to seek for the highest taxonomic resolution. In our opinion, the information that FISH provides is more comprehensive compared to sequencing-based methods: besides bioaerosol identification (as a function of FISH probe choice) and enumeration, it also enables the direct visualization of individual particles. Especially in the

field of bioaerosols, with limited knowledge about transport modes/mixing state and the potential influence of particles' size, shape, and surface structure on atmospheric processes, we believe that the visualization can be an advantage. Just as reviewer # 3 stated; „the ability to observe and enumerate assemblages of different composition (for example, assemblages that consist of only bacteria or assemblages that consist of eukaryotes and bacteria) is a clear strength of this technique. Figure 4H is a beautiful example."

To highlight this strength of the FISH approach we added a new figure (Figure 4) showing further microscopic images of Amazonian bioaerosols and included a description:

[Figure]

**Figure 4.** Microscopic images of fluorescence signals after DNA staining with DAPI (blue) and FISH (eukaryotes in orange and bacteria in red). Bioaerosol samples were collected at 5 m height. Except for one bacterial bioaerosol in panel C, all other fluorescent bioaerosol signals in these panels were attributed to the eukaryotic domain.

(P 10 L23-P11 L14): *Finally, the microscopic visualization of cells after staining also provides qualitative insights into the Amazonian bioaerosol population, which is a strength of the FISH-approach. Figure 4 shows selected examples of bioaerosols typically found at 5 m height at the ATTO site. Most of the bioaerosol visualized in Figure 4 belong to the eukaryotic domain. Some of them could also be identified as spores based on morphological criteria (Gregory et al., 1973; Lacey and West, 2007). Figure 4A further illustrates the importance of a careful*

*fixation and permeabilization prior to hybridization to enable the entrance of the FISH-probe into the cells. Here, the fern spores – identified by their typical spike-like surface structure according to Lacey and West (2007) – show nearly no orange fluorescence indicating a lack of hybridized eukaryotic probe due to insufficient cell lysis. In comparison, the ascospore in Figure 4B shows intense orange fluorescence as a sign of successful hybridization. However, signal intensities may vary also due to different rRNA contents as a matter of metabolic activity (positive signal but overall low fluorescent intensity of the spore on the left site in Figure 4C). Here, the manual microscopic inspection is of advantage as parameters such as particle size, morphology, surface structure, and fluorescent color can be considered beyond fluorescence intensity, to discriminate biological from non-biological and potentially autofluorescent particles. In terms of counting statistics, the manual enumeration can be beneficial as particles yielding two or more DAPI stained cores can be identified as single bioaerosol as shown in form of an ascospore (white arrows) in Figure 4D.*

Regarding the quantification of bioaerosols, we are convinced that FISH is a technique that provides robust concentrations of airborne cells for clearly defined taxonomic (or functional) groups on the basis of direct enumeration (instead of calculation like done for qPCR). Therefore, we purposefully decided to choose FISH probes to target the domain levels and to obtain number concentrations of Bacteria, Archaea, and Eukaryotes. To clarify this aspect, the following text has been added / section has been modified:

(P10, L23-24): Finally, the microscopic visualization of cells after staining also provides qualitative insights into the Amazonian bioaerosol population, *which is a strength of the FISH-approach.*

(P14, L2-12): Our study showed that FISH has great analytical potential in bioaerosol analysis. It combines bioaer-osol identification, quantification, *as well as visualization* and, thus, provides insights into the concentration levels and spatiotemporal variability of specific and clearly defined organism groups with-in the bioaerosol population. *Besides airborne abundances, only little is known about single particle properties such as identity, mixture or size. Here, we propose FISH to be a promising tool, to complement the methods currently established for environmental bioaerosol analysis (Sect. S1.4). As this is the first study using FISH for Amazonian bioaerosol analysis, we decided for three broad taxonomic probes, to create a first overview on domain level before exploring the bioaerosol population at a higher taxonomic resolution.* The Amazonian bioaerosols were investigated on domain level by quantifying eukaryotic, bacterial, and archaeal cells as well as the overall concentrations of airborne cells as a function of time and height within and above the forest canopy.

Further the reviewer stated "16S rRNA qPCR is more robust than microscopy for counting as well." We oppose the reviewer's position here. (i) Generally, nucleic acid staining is a widely used, well established and a rather simple approach for total cell quantification (e.g., Kepner and Pratt, 1994; Matthias-Maser and Jaenicke, 1995; Eduard and Heederik, 1998; Amato et al., 2007; Yamaguchi et al., 2012; Liu et al., 2019). Since a DAPI counter-staining (as well as a comparison to the aerosol online instruments) has been conducted here and yielded very consistent concentrations overall, we are confident that our FISH counts are robust. (ii) Further, we sought for a quantification of bioaerosols (on domain level) that also provides information on the *airborne state* of the cells. Thus, we chose a combination of sampling (membrane filters) and analysis through visualization (FISH with microscopy) that largely conserves airborne cluster of cells or agglomerates of cells and non-biological particles. For a sequencing-based approach (e.g. qPCR), information on cells clusters or agglomerates would be entirely lost after sampling (high-volume filter samplers or impingers), DNA extraction, and sequencing. (iii) Further, sequencing requires DNA extraction, either after resuspension from HiVol filters or directly from impinger samples (cell suspensions). In both cases, uncertainties remain on whether the DNA extraction occur quantitatively (due to thick cell walls and inequal cell lysis), which might bias the finally calculated cell number concentration. The choice of the extraction protocol might introduce an additional source of cell number underestimation and/or overestimation of certain clades. Gene copy number variations are still a challenge in the accurate estimation of microbial cell concentrations. In addition, the presence of eDNA can further complicate the quantification using qPCR, which might have a significant influence, especially when it comes to low-biomass aerosol samples (Luk et al., 2018).

Supplement, P4, L11- P6, L11:
*S1.4 Bioaerosol analysis methods*
*Environmental bioaerosol populations comprise highly complex and diverse particle mixtures. As a consequence, the choice of analysis method is not trivial and has to be made carefully to avoid biases caused by e.g., differences in particle size, metabolic state, or physical and chemical properties. A long time, cultivation was the method of choice for bioaerosol analysis. Since less than 1% of all bioaerosols are assumed to be culturable, this technique was more and more superseded by new analytical methods based on DNA analysis or real-time autofluorescence detection. To find the most suitable analysis technique, the main investigation target has to be defined first. We suggest the categorization into three objectives: 1. quantification, 2. identification, and 3. qualitative analysis of bioaerosols.*

*Online autofluorescence detectors such as the Rapid-E (Plair), BioScout (Environics, Ltd.) and spectral intensity bioaerosol sensor (Droplet Measurement Technologies) are especially useful for long-term quantification of bioaerosols, as no time-consuming laboratory sample analysis is needed. Furthermore, data is generated in comparably high time resolution. Nevertheless, data provided by these detectors has to be evaluated carefully, as bioaerosol number concentrations based on autofluorescence detection are prone to biases caused by bioaerosols' diverse autofluorescence intensities and wavelength range as well as interferences with autofluorescence from inorganic aerosols (e.g., Pöschl et al., 2010; Huffman et al., 2010; Savage and Huffman, 2018). In contrast, methods based on DNA analysis are focused on the specific identification of bioaerosols. By now, a broad range of tools are used, such as metagenome sequencing, metabarcoding, rRNA sequencing, or qPCR. The different techniques have in common that the taxonomic resolution is determined by the choice of the target sequence. Next to the taxonomic identification, microbial abundances can be quantified indirectly, e.g., by calculating microbial cell numbers out of detected gene copy numbers. These calculations are statistically robust, as DNA analysis requires high load of biological sample material on one hand, and automated high-throughput instruments enable a quick analysis of large sample numbers on the other hand. However, they can be biased by multiple copy numbers of marker genes. With the microscopic analysis of FISH, we combine the quantification and identification with a qualitative analysis. Even though the manual enumeration of fluorescent single particles is time consuming (automated counting can speed up the analysis), we suppose the advantages to outweigh this drawback: i) The quantification is based on direct enumeration of fluorescent cell signals and therefore assumed to be very accurate. ii) A countercheck with DAPI staining provides additional safety. iii) Particle loss during laboratory analysis is assumed to be minimized, as bioaerosols are identified directly on the filter as collection medium. For qPCR or flow cytometric analysis a re-suspension into liquid and a cell concentration is required after bioaerosol filtration, which enhances the chance of particle loss, especially in terms of charged and/or hydrophobic bioaerosols such as certain molds. Bioaerosol collection directly into liquid (e.g., impingement) could solve this issue. However, varying collection efficiency due to liquid evaporation over time, changes in chemical composition (e.g., pH or fixative concentration) as well as microbial growth within the liquid have to be taken into account.*

*Generally, it is important to note that the large bioaerosol diversity imposes significant analytical challenges in terms of sound bioaerosol analysis. There is no general "method of choice" for bioaerosol analysis, but various different approaches, that have advantages and*

*drawbacks. Accordingly, number and mass concentrations derived from different measurement techniques and sampling locations are comparable only within certain limits and similarities as well as deviations have to be evaluated carefully (see Table S3). We suppose that FISH, which was considered before but never established for environmental bioaerosol investigations, can advance the range of tools and techniques by combining the three major goals that are identification, enumerations and qualitative analysis.*

The manuscript lacks an explanation why sequencing techniques were not applicable for this system (why wasn't FISH and sequencing done?)

We absolutely agree that sequencing in parallel to the FISH sampling would have opened perspectives for interesting comparisons. The main reason why sampling for sequencing was not conducted in this study were the generally challenging field logistics and work at the ATTO site, which requires careful planning of all activities. This refers especially to activities on the tall tower. In this 'FISH-focused field campaign' in 2018 analyzed here, we put a clear focus on the implementation and testing of the FISH sampling protocol, using 3 sampling heights. Based on the interesting and encouraging results obtained here, we conducted a follow-up campaign in 2019 including filter sampling for sequencing, which is currently subject of in-depth analysis.

We added a reference that gives an overview about potential bioaerosol analysis techniques and linked the new supplement paragraphs in the main article:

(P3, L14-18): Particularly scarce are techniques that provide atmospheric number concentrations for specific and clearly defined organism groups within the bioaerosol population *(e.g., Kabir et al., 2020; Mbareche et al., 2017; Sect. S1.4).*

[R1.3] What is the standard microbial density of a given aggregate? Are all domains found in physical association? Separate? How are they dispersed on the aggregate? This would be something a sequencing-only study could not provide and better justify the methods used. There are qualitative statements to this effect that could be expanded upon (pg 10 lns 13-16).

[A1.3] It is not quite clear what the reviewers means with "microbial density of a given aggregate". We assume that this refers to degree of aggregation of the cells. If Reviewer #1 is asking for a ratio (such as cells per agglomerate) this is nothing we measured here. We complemented the paragraph in the article as follows:

(P12, L 811): *In the course of the microscopic analysis Archaea were found as single particles only. Fungal spores were occasionally found in physical association with bacteria (as shown*

*in Figure 4G and H) or with other fungal spores.* The vast majority of cells, however, was observed as separated cells, which suggests that under the given wet season conditions the bioaerosol components are largely externally mixed.

3) [R1.4] It is unclear why there is an improved understanding of bioaerosols from the number of bacteria present as opposed to total number of microbes present. For example, the paper states that these data provide constraints on mixing information, but it is not known if the bacteria are the same or different throughout. Is it the same population of bacteria that travel through the different heights? Or entirely different bacteria?

[A1.4] This study gives first insights into the general mixing of bioaerosols on domain level. However, these results don't allow assumptions about different types or populations of bacteria at different heights. The probe used here is applied for the general identification of bacteria to create an overview image.

In general, we expect that other bacterial probes, on higher taxonomic resolution, could answer the question of Reviewer #1. However, bacteria were found in relatively low abundance throughout this study. Accordingly, we expect too low bacterial numbers on the filters used here to be hybridized with a more specific probe but still ensuring robust statistics. As a result, the sampling volume would need to be increased in order to proceed with this approach.

**Minor and specific comments:**

[R1.5] Methods - It is not clear how the determination for the genome size of all bacteria and archaea was made. Was this an average of all the genomes? How might this vary between cells?

[A1.5] We have clarified this aspect and improved the paragraph "Quantification of atmospheric DNA concentration" in the appendix. Specifically, we included a link to the NCBI database that was used for the genome size determination.

To (P19, L13-19): *The genome sizes were determined as follows: Souza et al. (2019) found Proteobacteria and Actinobacteria to be the dominant phyla within the airborne Amazonian bacterial population. The median genome sizes found in the NCBI database were ~ 4.8 Mb and ~4.3 Mb for Proteobacteria and Actinobacteria, respectively. (https://www.ncbi.nlm.nih.gov/genome/browse#!/prokaryotes/proteobacteria). By discussing these numbers and the results by Landenmark et al., 2015 with ecologists from the Max Planck*

*Institute for marine Ecology in Bremen, the approximate bacterial and archaeal genome size were defined as 4 Mb for bioaerosols.*

[R1.6] The SD is very high compared to the average (Table 1). How many filters were counted? A supplemental table of each count conducted per time/height would be useful. It would be useful to know the variability in counting a given filter (variability in counts per field of view) separately from the deviation between filters counted for a given experimental filter (Table 1).

[A1.6] One filter was collected and counted per sampling height and sampling period. That results in 16 filters that were analyzed in total, five of those were collected at 5 m height and at 325 m height, respectively, and 6 were collected at 60 m height. These numbers as well as the FISH and DAPI raw counts per filter are presented in Table S1 in the supplement. Naturally, the SD is higher for samples with low numbers of raw counts, for example the SD for archaeal counts is higher than for eukaryotic counts In general, we assume the SD to be an acceptable range.

The parameter to ensure robust statistics was not the number of fields of view, but the number of DAPI stained particles that were enumerated (also given in table S1). To avoid confusion, we therefore refrain from presenting the number of fields of view (FOV) in an additional table. However, we understand the interest of Reviewer #1. The following numbers might give Reviewer #1 an impression regarding the number of FOV: at 5 m height on average 17 FOV were inspected, at 60 m height on average 28 FOV were inspected and at 325 m height on average 26 FOV were inspected, each per FISH probe. In total 1140 FOV were analyzed for the data presented in the study.

[R1.7] Man-made could be replaced with anthropogenic or human.

[A1.7] We replaced "man-made" by "human" and "anthropogenic" as suggested.

(P3, L22): unperturbed by *human* emissions

(P3, L27): the nowadays pervasive *anthropogenic* emissions and activities

[R1.8] Figure 2 – why are there no "unknowns" for Mar 1 and Mar 2?

FISH numbers are never a 100 % reflection of the reality but it can suffer from false negative but also false positive counts, as mentioned in the article. This can be due to different artefacts such as e.g. low signal intensity due to little number of bound fluorescent molecules and on the other hand also unspecific binding of probes or autofluorescent particles. For March 1 and March 2, we expect this to be the case. Especially archaeal numbers were comparably high at 60 m. The probe ARCH915 was reported to be prone to unspecific binding (Pernthaler et al.,

2002). We assume that this in combination with the low counting statistics for Archaea might be the reason for the "missing" unknowns during these two sampling days. We completed the respective footnote in the article, to point that our more clearly:

(P6, footnote):  In fact, we refrain from interpreting NARC in great detail in this work due to the low statistics. Furthermore, the probe ARCH915 used here was found to hybridize with some Bacteria, which could lead to false-positive signals. *We assume that this could have been the case on March 1 and March 2 at 60 m sampling height, leading to no "unknowns" in respect to DAPI numbers.*

References:

Amato, P., Parazols, M., Sancelme, M., Mailhot, G., Laj, P., and Delort, A.-M.: An important oceanic source of micro-organisms for cloud water at the Puy de Dôme (France), Atmospheric Environment, 41, 8253-8263, https://doi.org/10.1016/j.atmosenv.2007.06.022, 2007.

China, S., Burrows, S. M., Wang, B., Harder, T. H., Weis, J., Tanarhte, M., Rizzo, L. V., Brito, J., Cirino, G. G., Ma, P.-L., Cliff, J., Artaxo, P., Gilles, M. K., and Laskin, A.: Fungal spores as a source of sodium salt particles in the Amazon basin, Nature Communications, 9, 4793, 10.1038/s41467-018-07066-4, 2018.

Eduard, W., and Heederik, D.: Methods for quantitative assessment of airbone levels of noinfectious microorganisms in highly contaminated work environment, American Industrial Hygiene Association Journal, 59, 113-127, 10.1080/15428119891010370, 1998.

Elbert, W., Taylor, P. E., Andreae, M. O., and Pöschl, U.: Contribution of fungi to primary biogenic aerosols in the atmosphere: wet and dry discharged spores, carbohydrates, and inorganic ions, Atmospheric Chemistry and Physics, 7, 4569-4588, 2007.

Huffman, J. A., Treutlein, B., and Pöschl, U.: Fluorescent biological aerosol particle concentrations and size distributions measured with an Ultraviolet Aerodynamic Particle Sizer (UV-APS) in Central Europe, Atmospheric Chemistry and Physics, 10, 3215-3233, 2010.

Kabir, E., Azzouz, A., Raza, N., Bhardwaj, S. K., Kim, K.-H., Tabatabaei, M., and Kukkar, D.: Recent Advances in Monitoring, Sampling, and Sensing Techniques for Bioaerosols in the Atmosphere, ACS Sensors, 5, 1254-1267, 10.1021/acssensors.9b02585, 2020.

Kepner, R. L., and Pratt, J. R.: Use of fluorochromes for direct enumeration of total bacteria in environmental-samples - Past and present, Microbiological Reviews, 58, 603-615, 1994.

Liu, T., Chen, L. W. A., Zhang, M., Watson, J. G., Chow, J. C., Cao, J., Chen, N., Yao, R., Xiao, W., Chen, H., Wang, W., Zhang, J., Zhan, C., Liu, H., and Zheng, J.: Bioaerosol Concentrations and Size Distributions during the Autumn and Winter Seasons in an Industrial City of Central China, Aerosol and Air Quality Research, 19, 1095-1104, 10.4209/aaqr.2018.11.0422, 2019.

Luk, A. W., Beckmann, S., and Manefield, M.: Dependency of DNA extraction efficiency on cell concentration confounds molecular quantification of microorganisms in groundwater, FEMS microbiology ecology, 94, fiy146, 2018.

Matthias-Maser, S., and Jaenicke, R.: The size distribution of primary biological aerosol particles with radii >0.2 µm in an urban rural influenced region, Atmospheric Research, 39, 279-286, 1995.

Mbareche, H., Brisebois, E., Veillette, M., and Duchaine, C.: Bioaerosol sampling and detection methods based on molecular approaches: No pain no gain, Sci. Total Environ., 599, 2095-2104, 2017.

Moran-Zuloaga, D., Ditas, F., Walter, D., Saturno, J., Brito, J., Carbone, S., Chi, X., Hrabě de Angelis, I., Baars, H., Godoi, R. H. M., Heese, B., Holanda, B. A., Lavrič, J. V., Martin, S. T., Ming, J., Pöhlker, M. L., Ruckteschler, N., Su, H., Wang, Y., Wang, Q., Wang, Z., Weber, B., Wolff, S., Artaxo, P., Pöschl, U., Andreae, M. O., and Pöhlker, C.: Long-term study on coarse mode aerosols in the Amazon rain forest with the frequent intrusion of Saharan dust plumes, Atmos. Chem. Phys., 18, 10055-10088, 10.5194/acp-18-10055-2018, 2018.

Morris, C. E., Sands, D. C., Glaux, C., Samsatly, J., Asaad, S., Moukahel, A. R., Gonçalves, F. L. T., and Bigg, E. K.: Urediospores of rust fungi are ice nucleation active at > −10 °C and harbor ice nucleation active bacteria, Atmos. Chem. Phys., 13, 4223-4233, 10.5194/acp-13-4223-2013, 2013.

Pernthaler, A., Preston, C. M., Pernthaler, J., DeLong, E. F., and Amann, R.: Comparison of Fluorescently Labeled Oligonucleotide and Polynucleotide Probes for the Detection of Pelagic Marine Bacteria and Archaea, Applied and Environmental Microbiology, 68, 661-667, 10.1128/aem.68.2.661-667.2002, 2002.

Pöhlker, C., Wiedemann, K. T., Sinha, B., Shiraiwa, M., Gunthe, S. S., Smith, M., Su, H., Artaxo, P., Chen, Q., Cheng, Y. F., Elbert, W., Gilles, M. K., Kilcoyne, A. L. D., Moffet, R. C., Weigand, M., Martin, S. T., Pöschl, U., and Andreae, M. O.: Biogenic Potassium Salt Particles as Seeds for Secondary Organic Aerosol in the Amazon, Science, 337, 1075-1078, 10.1126/science.1223264, 2012.

Pöschl, U., Martin, S. T., Sinha, B., Chen, Q., Gunthe, S. S., Huffman, J. A., Borrmann, S., Farmer, D. K., Garland, R. M., Helas, G., Jimenez, J. L., King, S. M., Manzi, A., Mikhailov, E., Pauliquevis, T., Petters, M. D., Prenni, A. J., Roldin, P., Rose, D., Schneider, J., Su, H., Zorn, S. R., Artaxo, P., and Andreae, M. O.: Rainforest Aerosols as Biogenic Nuclei of Clouds and Precipitation in the Amazon, Science, 329, 1513-1516, 10.1126/science.1191056, 2010.

Prenni, A. J., Petters, M. D., Kreidenweis, S. M., Heald, C. L., Martin, S. T., Artaxo, P., Garland, R. M., Wollny, A. G., and Pöschl, U.: Relative roles of biogenic emissions and Saharan dust as ice nuclei in the Amazon basin, Nature Geoscience, 2, 401-404, 10.1038/ngeo517, 2009.

Prospero, J. M., Blades, E., Mathison, G., and Naidu, R.: Interhemispheric transport of viable fungi and bacteria from Africa to the Caribbean with soil dust, Aerobiologia, 21, 1-19, 2005.

Savage, N. J., and Huffman, J. A.: Evaluation of a hierarchical agglomerative clustering method applied to WIBS laboratory data for improved discrimination of biological particles by comparing data preparation techniques, Atmos. Meas. Tech., 11, 4929-4942, 2018.

Souza, F. F., Rissi, D. V., Pedrosa, F. O., Souza, E. M., Baura, V. A., Monteiro, R. A., Balsanelli, E., Cruz, L. M., Souza, R. A., and Andreae, M. O.: Uncovering prokaryotic biodiversity within aerosols of the pristine Amazon forest, Sci. Total Environ., 688, 83-86, 2019.

Womack, A. M., Artaxo, P. E., Ishida, F. Y., Mueller, R. C., Saleska, S. R., Wiedemann, K. T., Bohannan, B. J. M., and Green, J. L.: Characterization of active and total fungal communities in the atmosphere over the Amazon rainforest, Biogeosciences, 12, 6337-6349, 10.5194/bg-12-6337-2015, 2015.

Yamaguchi, N., Ichijo, T., Sakotani, A., Baba, T., and Nasu, M.: Global dispersion of bacterial cells on Asian dust, Scientific Reports, 2, 525, 10.1038/srep00525, 2012.

---

## Author Comment (AC2)

**Manuscript format description:**

Black text shows the original referee comment, blue text shows the authors response, and red text shows quoted manuscript text. Changes to the manuscript text are shown as *italicized and underlined*. We used bracketed comment numbers for referee comments (e.g., [R1.1]) and author's responses (e.g., [A1.1]). Line numbers refer to the discussion/review manuscript.

**Anonymous Referee #2,**

Received: 19 Feb 2021

 M. Prass et al. present a paper entitled « Bioaerosols in the Amazon rain forest: Temporal variations and vertical profiles of Eukarya, Bacteria and Archaea". They measured the concentrations of these types of bioaerosols during a campaign of six days (sampling every day) at three levels of altitude (5, 30 and 325 m) on the Amazon Tall Tower Observatory (ATTO) during typical wet season conditions.

The obtained results are of interest as the scientific community needs data on bioaerosol concentrations in various sites and under defined atmospheric conditions in order to understand the transport and potential roles of bioaerosols in atmospheric processes (cloud chemistry and microphysics, precipitation formation…). However, I have some major concerns about the introduction, discussion and conclusion of the paper, these parts of the paper should be partly rewritten before the paper is published.

We thank Referee #2 for the evaluation of our work. We appreciate the critical and constructive analysis and answered the comments in detail below:

**Major comments:**

[R2.1] Abstract line 5-6, Introduction P3 lines 30-37, P4 lines 1-3, Conclusion P12 1-5, 32 (etc..): The authors focus the interest of their paper on the use of the FISH method as an "analytical novelty "and a great improvement to analyze and quantify bioaerosols.

First, this method is not new and was used, even on bioaerosols, quite a long time ago (see for instance Lange et al. Application of Flow Cytometry and Fluorescent In Situ Hybridization for

Assessment of Exposures to Airborne Bacteria, Appl. Environ. Microbiol. 1997, 63, 1557–1563).

[A2.1] We appreciate that the reviewer questions whether the scientific novelty might be overstated. We checked the manuscript – and especially the sections that the reviewer pointed out – critically again. We came to the conclusions that these statements are generally appropriate and not overstated. Nevertheless, we found room for further clarification and revised the text accordingly as outlined below. Specifically, the reviewer refers to the following statement here:

(i)   Abstract, L5-6: We used fluorescence *in situ* hybridization (FISH), a molecular biological staining technique largely unexplored in aerosol research, to investigate the sources and spatiotemporal distribution of Amazonian bioaerosols on domain level.

(ii)  P3, L31-37 & P4, L1-3: The analytical and scientific novelty of this study is threefold: First, it widens the spectrum of techniques for bioaerosol investigations by exploring the analytical potential of fluorescence in situ hybridization (FISH) in this field. FISH is a molecular genetic technique for the specific staining of cells by targeting characteristic RNA or DNA sequences with complementary and fluorescently labeled nucleotide probes (e.g. Amann and Fuchs, 2008). In terrestrial and marine microbiology, FISH has become an important technique in identification and enumeration of microbial organisms with numerous applications (e.g., Pernthaler et al., 2004; Christensen et al., 1999). However, applications in bioaerosol research have remained remarkably sparse (e.g., Lange J. L., 1997; Yoo et al., 2017; Harrison et al., 2005). Our results demonstrate that FISH has great potential in bioaerosol analysis as it provides number concentrations of specific organism classes (i.e., from domain down to species level) and, therefore, combines *bioaerosol identification, enumeration, and visualization.*

(iii) P14, L1-5: Our study showed that FISH has great analytical potential in bioaerosol analysis. It combines bioaerosol identification, quantification, *as well as visualization* and, thus, provides insights into the concentration levels and spatiotemporal variability of specific and clearly defined organism groups within the bioaerosol population.

We do not see that these sections would claim that FISH is new. In fact, several references – also two studies on previous FISH applications on bioaerosols (Yoo et al., 2017 and Harrison et al., 2005) are cited. The number of FISH studies on bioaerosols is very small, however, which is addressed by saying that the combination of FISH and bioaerosols is largely unexplored. In terms of novelty of this study we state that the study widens the spectrum of techniques for bioaerosol investigations by exploring the analytical potential of fluorescence in situ hybridization (FISH) in this field, which seems not to claim that we "introduced" or "developed" FISH for bioaerosol analysis here.

The reference (Lange et al., 1997) that Reviewer #2 mentioned has been added on P43, L1. We would like to point out here that all previous studies that we are aware of and that used FISH for bioaerosol analysis did not focus on bioaerosol samples from natural habitats. Instead they addressed laboratory-generated bioaerosols as well as 'artificial' environments such as swine bars. This is part of the reason for our statement that this field of applications is largely unexplored. Clearly, the bioaerosol populations in e.g. swine bars and the Amazonian atmosphere are quite different (e.g., diversity, concentration, emission patterns, etc.). Accordingly, an achievement and novelty of our work was the adaptation of existing FISH protocols (e.g., fixation, permeabilization and hybridization) to the highly diverse mixture of atmospheric bioaerosols, which has been outlined transparently in the manuscript as detailed experimental protocols for further use by other researchers.

Along the lines of [R2.1] and [A2.1], we have implemented the following changes in the manuscript for further clarification:

We added the reference Referee #2 mentioned in:
(P3 L32- P4, L2): The analytical and scientific novelty of this study is threefold: First, it widens the spectrum of techniques for bioaerosol investigations *in environmental samples* by exploring the analytical potential of fluorescence *in situ* hybridization (FISH) in this field. FISH is a molecular genetic technique for the specific staining of cells by targeting characteristic RNA or DNA sequences with complementary and fluorescently labeled nucleotide probes (e.g., Amann and Fuchs, 2008). In terrestrial and marine microbiology, FISH has become an important technique in identification and enumeration of microbial organisms with numerous applications (e.g., Pernthaler et al., 2004; Christensen et al., 1999). However, applications in

bioaerosol research have remained remarkably sparse (e.g., Lange et al., 1997; Yoo et al., 2017; Harrison et al., 2005)

[R2.2] Second, although this method is valuable to distinguish different types of bioaerosols using specific probes (bacteria, eukaryotes and archaea, or different genera, species, etc….), it is a very laborious and time consuming approach when combined with microscopy, in addition it may presents some artifacts and so the quantification of bioaerosols must be first validated using another well-established method.

This is actually recognized by the authors (P 4 lines 19-23). In this paper they used DAPI staining combined with microscopic observation to quantify the total number of cells and validate their data. Also this method is very laborious and time consuming. DAPI and FISH combined with microscopic observations require a great number of counts to avoid errors and are also dependent on the observer faculties, as explained p16 lines 11-15, 31-35.

[A2.2] DAPI staining combined with microscopic detection of fluorescent signal is a well-established method which was used in studies for the quantification of bioaerosols before. Therefore, we indeed regard these numbers as a valuable and robust reference for the FISH counts. Furthermore negative tests were performed by analyzing blanks as well as by the application of the non-specific probe. Positive test by applying the bacterial probe to filters covered with *E.coli* were performed to ensure the hybridization with the fluorescent probes was successfull. In addition, the total aerosol counts in the same size range by the OPS give a statistic robust estimation of (bio-)aerosol number concentrations.

The statistical evidence of the FISH and DAPI counts as well were achieved by counting not less than 500 particles, a number that was established by microbiologists using the method for marine science for many years now (Kirchman et al., 1982; Pernthaler et al., 2003).

[R2.3] I do recognize the efforts of the authors and the serious of their job, that finally gave valuable data. However, we cannot consider this method as the "future of the bioaerosol characterization and quantification" as there are alternative methods which are now currently used.

[A2.3] As already explained as answer to comment from Referee # 1 [A1.2],

we wouldn't dedicate FISH as "the only future of bioaerosol analysis", but [P3, L31] it widens the spectrum of techniques for bioaerosol investigations in environmental samples by a visualization and localization tool.

(P15, L3-6): Finally, the taxonomic resolution of this study operates exclusively on domain level. Future studies should use the analytical potential of FISH by targeting organism classes on lower taxonomic levels (e.g., theoretically down to species level) *in combination with sequencing-based techniques.*

(P14, L32-34): For this purpose, a broader statistical basis of FISH results and comparisons with bioaerosol analysis techniques *(such as sequencing or qPCR)* along with meteorological observations is needed.

[R2.4] The authors could refer to a recent review by Kabir et al. (Recent Advances in Monitoring, Sampling, and Sensing Techniques for Bioaerosols in the Atmosphere, ACS Sens. 2020, 5, 1254−1267). In my opinion two main approaches can be interesting:

1- Flow Cytometry: this method is not new (see Lange et al 1997 cited above) but is very efficient, cheap and quick, in addition improvements of flow cytometers open new possibilities. Many samples can be analyzed; it allows to quantify the total number of cells but also to distinguish different types of cells using specific probes (bacteria, fungal spores, genera ….), or dead and alive cells using viability markers, or photosynthetic cells using specific wavelengths of emission of these organisms. This analyze can be performed directly on atmospheric samples with intact cells in water medium (rain, snow, aerosols in impingers, resuspension of aerosols from filters…). See for instance some papers:

Chena and Li. Real-time monitoring for bioaerosols—flow cytometry, Analyst, 2007, 132, 14–16, DOI: 10.1039/b603611m.

Liang et al., Rapid detection and quantification of fungal spores in the urban atmosphere by flow cytometry, J. Aerosol Sci. 2013,66, 179–186.

Jang et al., Application of Cytosense flow cytometer for the analysis of airborne bacteria collected by a high volume impingement sampler, J. Microbiol. Methods, 2018, 154, 63-72.

Negron et al. Using flow cytometry and light-induced fluorescence to characterize the variability and characteristics of bioaerosols in springtime in Metro Atlanta, Georgia. Atmos. Chem. Phys., 2020, 20, 1817–1838, https://doi.org/10.5194/acp-20-1817-2020.

Dillon et al., Cyanobacteria and Algae in Clouds and Rain in the Area of puy de Dôme, Central France, Appl. Environ. Microbiol. ,2020 , 87 , e01850-20. doi: 10.1128/AEM.01850-20

2- qPCR: This molecular approach is more recent but has been used intensively in recent papers, it allows all types of quantifications (total number of cells, specific counts…). This approach is also quick, cheap and easy to run. See for instance (among many others):

Tignat-Perrier et al., Global airborne microbial communities controlled by surrounding landscapes and wind conditions, Sci. Rep., 2019, 9, 14441 , https://doi.org/10.1038/s41598-019-51073-4.

Some papers combine flow cytometry and qPCR, and even new-generation sequencing approaches (metagenomics), see for instance (among others):

Dillon et al., Cyanobacteria and Algae in Clouds and Rain in the Area of puy de Dôme, Central France, Appl. Environ. Microbiol., 2020 , 87 , e01850-20. doi: 10.1128/AEM.01850-20.

[A2.4] We thank Referee #2 for the given reference, it was added in the introduction:

(P3, L14-16): Particularly scarce are techniques that provide atmospheric number concentrations for specific and clearly defined organism groups within the bioaerosol population (e.g., *Mbareche et al., 2017; Kabir et al., 2020; Sect. S1.4*).

We agree, flow cytometry and qPCR hold several advantages that make them suitable especially for quantification of bioaerosols in large sample sets. However, one of this study's scopes was the exploration of FISH for environmental bioaerosol analysis. We particularly supposed the visualization to be a promising tool, to enable the investigation of bioaerosol travel modes (e.g., agglomeration), a characteristic that cannot be approached by flow cytometry or qPCR.

 Comparisons of different analysis technique can be found in the literature, e.g., Ghosh et al. (2015) summarized a handy table providing basic advantages and drawbacks of different techniques. Nevertheless, we took R2.4 as a motivation to carefully compare FISH, qPCR, and flow cytometry ourselves during the review process. Please, find attached a table summarizing the method characteristics.

Another reason FISH seemed promising for us was the convincing sampling technique: With this study, we aimed to determine the characteristic cross section of a diel bioaerosol population during the wet season. Accordingly, sampling periods of approximately 24 hours were striven for. The dry collection by direct filtration onto polycarbonate membranes was our collection

method of choice as it minimizes the chance of unwanted microbial growth on or within the collection medium during that relatively long sampling time, as it would be expectable when sampling into liquids. Additionally, particles can be analyzed on the filters directly. A re-suspension and concentration by centrifuging, as required for sequencing or flow cytometric analysis, enhances the chance of particle loss, especially in terms of charged and/or hydrophobic bioaerosols such as certain molds (Mbareche et al., 2019).

[R2.5] To conclude, I propose the authors rewrite parts of their manuscript taking into account these alternative approaches, that I think are the future instead of FISH. They will allow to analyze many more samples in a much shorter time.
Actually it is also a weak point of this paper to have only a six-day sampling period, with three altitudes, instead of larger series of data that could bring more information and robust statistical analyses. The obtained data are interesting, valuable but very limited.

[A2.5] Please, find the answer to the comment [R2.5] in [A3.2].

[R2.6] Another general comment is that lots of data were collected in this Amazon campaign thanks to this very well equipped observatory site (ATTO) but they were not really exploited to explain the observed trends concerning bio-aerosols quantification and distribution. Again this is due to the too small number of samples, the discussion part is thus very speculative.

[A2.6] We agree that the comprehensive data collected at the ATTO site on long-term basis is very valuable for the interpretation of targeted bioaerosol studies, such as the one presented in this manuscript. Evidently, some measurements are more closely related to the results here and thus have greater value for the interpretation than others. Here, selected and related data sets from online aerosol instruments were implemented that provide a valuable context for the pilot project collecting DAPI and FISH results. Specifically, data on aerosol concentration and size distributions (from SMPS and OPS instruments) were used (see. Fig. 2 and 3). Meteorological data are part of Fig. 2. The sampling period is embedded in the long-term perspective of climatic anomalies in Fig. S2. Air mass movement and through backward trajectory analysis is presented in Fig. S1. The reviewer is right, that the comparatively short length of the sampling period does not allow long-term correlations with e.g. meteorological parameters (e.g. bioaerosol release as a function of RH). Still, the comprehensive data context from ATTO allowed us to draw some general conclusions that we do not regard as speculative.

**Minor and specific comments:**

[R2.7] P3 lines 16-20: the cited literature is too limited; it has to be completed.

[A2.7] We cite three review papers at this point, to avoid naming only few field studies in the broad field of valuable work that has been done. To clarify that further references can be found in the review papers, we added the following appendix:

(P3, L17-19): The number of bioaerosol field observations worldwide is constantly increasing (Després et al., 2012; Fröhlich-Nowoisky et al., 2016; Šantl-Temkiv et al., 2019, *and references therein*) with bioaerosol studies in regions that are essential for the climate system being particularly relevant.

[R2.8] P8 line 2-8: It is important to assess the viability of the cells but the method used by the authors is not very accurate because a cell can be "intact" from a microscopic point of view but not alive or active. Alternative "live and dead cell" assay is much more adapted as well as ATP (adenosine triphosphate) or CTC (5-cyano-2,3-ditolyl tetrazolium chloride) assays. Could the authors comment this point?

[A2.8] One of the three scopes of this study was the elaboration of the FISH method's potential in bioaerosol analysis. The analysis provided number concentration of potentially intact cells, which in combination with the total aerosol counts could be used to estimate the fraction of fragments. It is true, viability is important in the broader picture of bioaerosol analysis, but it was beyond the focus of this study.

Furthermore, filter samples collected in the Amazon had to be fixed immediately after sampling to avoid microbial growth that biases the concentrations and to enable the entrance of the fluorescent dye and fluorescently labeled nucleotide probes. The fixation is of high importance, especially because samples were collected in a remote collection, causing long travel times when samples are temporarily stored on ice instead of -20°C. Ergo, in the course of DAPI and FISH analysis, the cells were per se dead.

[R2.9] Figure 4 G and H: the authors show the presence of agglomerates composed of clusters of bacteria and fungal spores in samples taken as the lower altitude. They assume that these aggregates are due to their higher sedimentation tendency and lower atmospheric residence times. This can be one of the hypotheses, but it could also be due to the closer presence of ground and trees, source of dust (from the soil) or biofilm agglomerates (from the phyllosphere). Your opinion?

[A2.9] We agree, this is an important information that should be included here:

(P12, L15-17): An enrichment of larger particles at the lower heights *is probably a result of the large number of emission sources underneath the canopy in combination with* higher sedimentation tendency and lower atmospheric residence times of large particles.

References:

Amann, R., and Fuchs, B. M.: Single-cell identification in microbial communities by improved fluorescence in situ hybridization techniques, Nat. Rev. Microbiol., 6, 339-348, 2008.

Christensen, H., Hansen, M., and Sørensen, J.: Counting and size classification of active soil bacteria by fluorescence in situ hybridization with an rRNA oligonucleotide probe, Appl. Environ. Microbiol., 65, 1753-1761, 1999.

Després, V. R., Huffman, J. A., Burrows, S. M., Hoose, C., Safatov, A. S., Buryak, G., Fröhlich-Nowoisky, J., Elbert, W., Andreae, M. O., Pöschl, U., and Jaenicke, R.: Primary biological aerosol particles in the atmosphere: a review, Tellus B, 64, 1-58, 10.3402/tellusb.v64i0.15598, 2012.

Fröhlich-Nowoisky, J., Kampf, C. J., Weber, B., Huffman, J. A., Pöhlker, C., Andreae, M. O., Lang-Yona, N., Burrows, S. M., Gunthe, S. S., and Elbert, W., , Su, H., Hoor, P., Thines, E., Hoffmann, T., Després, V. R., and Pöschl, U.: Bioaerosols in the Earth system: Climate, health, and ecosystem interactions, Atmospheric Research, 182, 346-376, 2016.

Ghosh, B., Lal, H., and Srivastava, A.: Review of bioaerosols in indoor environment with special reference to sampling, analysis and control mechanisms, Environment International, 85, 254-272, https://doi.org/10.1016/j.envint.2015.09.018, 2015.

Harrison, R. M., Jones, A. M., Biggins, P. D. E., Pomeroy, N., Cox, C. S., Kidd, S. P., Hobman, J. L., Brown, N. L., and Beswick, A.: Climate factors influencing bacterial count in background air samples, Int. J. Biometeorol., 49, 167-178, 10.1007/s00484-004-0225-3, 2005.

Kabir, E., Azzouz, A., Raza, N., Bhardwaj, S. K., Kim, K.-H., Tabatabaei, M., and Kukkar, D.: Recent Advances in Monitoring, Sampling, and Sensing Techniques for Bioaerosols in the Atmosphere, ACS Sensors, 5, 1254-1267, 10.1021/acssensors.9b02585, 2020.

Kirchman, D., Sigda, J., Kapuscinski, R., and Mitchell, R.: Statistical analysis of the direct count method for enumerating bacteria, Applied and Environmental Microbiology, 44, 376-382, 1982.

Lange J. L., T. P. S., Lynch N.: Application of flow cytometry and fluorescent in situ hybridization for assesment of exposures to airborne bacteria, Appl. Environ. Microbiol. 1997, 63(4):1557., 1997.

Mbareche, H., Brisebois, E., Veillette, M., and Duchaine, C.: Bioaerosol sampling and detection methods based on molecular approaches: No pain no gain, Sci. Total Environ., 599, 2095-2104, 2017.

Mbareche, H., Morawska, L., and Duchaine, C.: On the interpretation of bioaerosol exposure measurements and impacts on health, Journal of the Air & Waste Management Association, 69, 789-804, 2019.

Pernthaler, A., Pernthaler, J., Amann, R., Kowalchuk, G., de Bruijn, F., Head, I., Akkermans, A., and van Elsas, J.: Sensitive multi-color fluorescence in situ hybridization for the identification of environmental microorganisms, Molecular microbial ecology manual. Volumes 1 and 2, 711-725, 2004.

Pernthaler, J., Pernthaler, A., and Amann, R.: Automated enumeration of groups of marine picoplankton after fluorescence in situ hybridization, Appl. Environ. Microbiol., 69, 2631-2637, 2003.

Šantl-Temkiv, T., Sikoparija, B., Maki, T., Carotenuto, F., Amato, P., Yao, M., Morris, C. E., Schnell, R., Jaenicke, R., Pöhlker, C., DeMott, P. J., Hill, T. C. J., and Huffman, J. A.: Bioaerosol Field Measurements: Challenges and Perspectives in Outdoor Studies, Aerosol Science and Technology, 1-41, 10.1080/02786826.2019.1676395, 2019.

Yoo, K., Lee, T. K., Choi, E. J., Yang, J., Shukla, S. K., Hwang, S.-i., and Park, J.: Molecular approaches for the detection and monitoring of microbial communities in bioaerosols: A review, Journal of Environmental Sciences, 51, 234-247, https://doi.org/10.1016/j.jes.2016.07.002, 2017.

---

## Author Comment (AC3)

**Manuscript format description:**

Black text shows the original referee comment, blue text shows the authors response, and red text shows quoted manuscript text. Changes to the manuscript text are shown as *italicized and underlined*. We used bracketed comment numbers for referee comments (e.g., [R1.1]) and author's responses (e.g., [A1.1]). Line numbers refer to the discussion/review manuscript.

**Anonymous Referee #3,**

Received: 25 Feb 2021

The manuscript by Prass and colleagues investigates the altitude distribution of bioaerosols in the Amazon rain forest. In summary, there are several strengths to this study: 1. bacteria, archaea, and eukarya were enumerated using fluorescent microscopy and FISH, 2. the site is an undisturbed forest environment, 3. collection was done at several different heights. There are also several weaknesses to this study: 1. only one week of data collection, 2. only one site of collection, 3. only one time/day of bioaerosol collection for FISH analysis, 4. Only one analytical technique (FISH) used to identify particles as bacteria, archaea, or eukarya.

We thank Reviewer #3 for their positive and critical evaluation. The reviewer's comments were very helpful to clarify several statements and arguments.

[R3.1] While FISH is a laborious technique and it has been used before on some aerosols to distinguish bacterial species from each other, it has not been used for atmospheric bioaerosols and not to distinguish bacteria, archaea, or eukarya from each other. In particular, the ability to observe and enumerate assemblages of different composition (for example, assemblages that consist of only bacteria or assemblages that consist of eukarya and bacteria) is a clear strength of this technique. Figure 4H is a beautiful example.

[A3.1] We thank Reviewer #3 for this positive overall evaluation.

[R3.2] The determination of absolute and relative numbers of bacteria, archaea, or eukarya at different heights in the Amazon rain forest is by itself another important result. However, it is clearly limited by the fact that collection was restricted to a one-time seven-day

period and that the results were not compared with any other analytical technique. Because FISH was not used before to determine concentrations of bacteria, archaea, or eukarya at different heights at other sites, the results cannot be compared with other studies that used different techniques.

[A3.2] We agree that the six-day sampling period (covering 3 sampling heights) analyzed here is comparatively short. Note, however, that more samples were collected in this sampling campaign and that the six days were analyzed on purpose as this time window had ideal conditions to investigate pristine atmospheric conditions in the Amazon (P4, L15-17, Supplement P2, L9-26, Figures S1, S2). Pristine episodes in the central Amazonian wet season are relatively rare and have an episodic character (for definition and details, see Pöhlker et al., 2018). Further note, that the sampling and analysis were embedded in a broad set of other meteorological, trace gas, and aerosol measurements running continuously at ATTO (e.g., (Andreae et al., 2015), which provided us a comprehensive context of data and, thus, allowed us to choose particularly interesting periods. In this sense, the present work can be regarded as a targeted case study on the bioaerosol population in the pristine Amazonian wet season as well as a proof of concept of the feasibility of our sampling and FISH protocols under challenging field conditions. Clearly, the results presented here spark a variety of follow-up questions (e.g., higher temporal resolution, day vs night differences, seasonal variations), which will be addressed in ongoing or planned follow-up campaigns. To emphasize these aspects more clearly in the manuscript, we rearranged and added the following statements in the main text:

(P4; L15-18) Revised: The samples analyzed in this study were collected during prevailing clean wet season conditions in the Amazon. *The six-day sampling period was chosen for detailed analysis as the aerosol mixture approximated a pre-industrial state with the bioaerosol population originating from the primary rain forest region within the ATTO site's footprint. A detailed characterization of the conditions can be found in the supplement.*

(P14, L10-14) The Amazonian bioaerosols were investigated on domain level by quantifying eukaryotic, bacterial, and archaeal cells as well as the overall concentrations of airborne cells as a function of time and height within and above the forest canopy. *These bioaerosol abundances are characteristic for naturally and clean background aerosol conditions as during the analyzed sampling period local emissions from the primary rain forest dominated.*

Regarding the reviewer's statements that "results were not compared with any other analytical technique" and that "results cannot be compared with other studies that used different techniques":

We agree here, though would like to point out a few aspects. Studies with quantitative bioaerosol analyses – especially in the Amazon – are very sparse. In this sense options for comparison with previous studies are inherently limited. Nevertheless, we conducted a literature synthesis and comparison with previous studies as far as possible by collecting all published number concentrations for biological aerosols (e.g., from microscopy and autofluorescence detection) in tropical (and boreal) forests in Table S3. We found that these results are largely consistent with our findings. Note in this context that the authors regard the uncertainties involved in autofluorescence-based (bio)aerosol detection as much larger as the uncertainties of the FISH approach. The agreement between Table S3 and the DAPI and FISH results made us confident that our data is a robust first data set e.g. on bacterial number concentrations in the Amazonian atmosphere.

Further, we conducted a careful comparison with online aerosol data and especially the data from an optical particle sizer, which provides a reference for the overall aerosol abundance in the DAPI and FISH relevant size range. We found a good agreement here as well. In fact, we consider consistent results with the overall aerosol variability (size distributions and total coarse mode concentrations) as equally important than a consistency for other molecular biological techniques as this is key for a comprehensive understanding of atmospheric processes at the interfaces of atmospheric physics, chemistry, and biology. The reviewer's comment made us aware that aforementioned aspects are probably not as clear as necessary in the manuscript in its current form. Thus, we revised the text accordingly and changes are specified under [A3.3].

[R3.3] Of course, it is impossible to change the study itself at this point. It is the opinion of this reviewer that some improvements to the manuscript itself can make this an important and interesting contribution. The main recommendation is to more clearly acknowledge the limitations I listed above. In particular, the authors should not state at the same time that their data provide "unprecedented insights" and are "highly consistent with … previous studies". The authors should instead acknowledge that the absence of an independent verification using

other techniques, such as sequence-based techniques or qPCR, and the absence of similar studies performed at other sites during other season limits the ability to compare their results and verify the accuracy of their results.

[A3.3] We appreciate this constructive recommendation and changed several statements in the manuscript accordingly. As described under A3.2, we agree that the comparability of this data set is partly hampered by the absence of studies from either the same location or studies using the same method. Nevertheless, we found bioaerosol number concentrations in tropical rainforest reported in a comparable range. We pointed out these limitations as follows:

Supplement, P6, L6-8): Accordingly, number and mass concentrations derived from different measurement techniques and sampling locations are comparable only within certain limits and similarities as well as deviations have to be evaluated carefully (see Table S3).

(P 2, L14-16) Before: The observed concentrations and profiles provide unprecedented insights into the sources and dispersion of different types of Amazonian bioaerosols as a solid basis for model studies on biosphere-atmosphere interactions such as bioprecipitation cycling. (P 2, L14-16) Revised: The observed concentrations and profiles provide new insights into the sources and dispersion of different types of Amazonian bioaerosols as a solid basis for model studies on biosphere-atmosphere interactions such as bioprecipitation cycling.

(P 4, L22-30) Revised: A focal point of this study has been the careful cross-validation and comparison of the obtained FISH results with online aerosol data as well as a synthesis with existing literature knowledge. This validation is important since FISH is experimentally demanding and prone to various artifacts (i.e. false positive or false negative counts) and thus may yield biased results (Thiele et al, 2011). _A comparison with data from different locations or obtained by different methodologies is meaningful only within certain limits (S1.4). Though, we overall_ found a high consistency with complementary online data from the ATTO site as well as from previous studies, which underlines that the obtained organism concentrations are a solid representation of the Amazonian wet season bioaerosol population.

(P 12, L25-28) Before: Overall, the results of this study greatly extend the knowledge on the life cycle of the Amazonian aerosols and provide a solid experimental basis for model investigations of bioaerosol-related processes, such as the role of biological ice nuclei or giant cloud condensation nuclei in cloud microphysics and potential bio-precipitation cycling.

(P 14, L24-27) Revised: Overall, the results of this study extend the knowledge on the life cycle of the Amazonian aerosols and provide a solid experimental basis for model investigations of bioaerosol-related processes, such as the role of biological ice nuclei or giant cloud condensation nuclei in cloud microphysics and potential bio-precipitation cycling.

(P 14, L33-34): For this purpose, a broader statistical basis of FISH results and comparisons *with bioaerosol analysis techniques (such as sequencing or qPCR)* along with meteorological observations is needed.

 (P15, L4-6): Future studies should use the analytical potential of FISH by targeting organism classes on lower taxonomic levels (e.g., theoretically down to species level) *in combination with and* *on the basis of sequencing-based techniques.*

References:

Andreae, M., Acevedo, O., Araùjo, A., Artaxo, P., Barbosa, C., Barbosa, H., Brito, J., Carbone, S., Chi, X., and Cintra, B.: The Amazon Tall Tower Observatory (ATTO): overview of pilot measurements on ecosystem ecology, meteorology, trace gases, and aerosols, Atmospheric Chemistry and Physics, 15, 10723-10776, 2015.

Pöhlker, M. L., Ditas, F., Saturno, J., Klimach, T., Hraběde Angelis, I., Araùjo, A. C., Brito, J., Carbone, S., Cheng, Y., and Chi, X.: Long-term observations of cloud condensation nuclei over the Amazon rain forest–Part 2: Variability and characteristics of biomass burning, long-range transport, and pristine rain forest aerosols, Atmospheric Chemistry and Physics, 18, 10289-10331, 2018.

---

## Referee Report (RR1)

We thank the authors for their effort to answer the concerns of the various reviewers and for the corrections of the manuscript. I agree it could be published but I would like the authors to make a few changes to take into account my previous comments.

P3 Line 36 : "Our results demonstrate that FISH has great potential in bioaerosol analysis as it provides number concentrations of specific organism classes (i.e., from domain down to species level) and, therefore, combines bioaerosol identification, enumeration, an visualization" I suggest to delete "great" and keep "FISH has potential in bioaerosol analysis". Indeed we think that this technique is so heavy that its use to analyze a great number of samples will be limited, alternative techniques will be more suitable (qPCR, sequencing, Flow cytometry + targeted probes etc..)

P 11 line 7: "Our study showed that FISH has great analytical potential in bioaerosol analysis." Please delete "great" (for the same reason as before).

P11 line 11: "Here, we propose FISH to be a promising tool". Please change "promising" for "interesting" (same reason).

*Supplement*

P 5 line: The authors should add a paragraph about the advantages of flow cytometry to quantify the total number of cells and look at their size distribution. This technique is very fast contrary to DAPI which is time consuming. They should also speak about the combination of flow cytometry with specific staining with targeted probes.

P5 line 23. We do not agree with this paragraph :impingement is recognized as an efficient tool and no growth is observed with a short time collection, typically less than an hour is need to have enough sample , especially using high volume impingers (Šantl-Temkiv, T., Sikoparija, B., Maki, T., Carotenuto, F., Amato, P., Yao, M., Morris, C. E., Schnell, R., Jaenicke, R., Pöhlker, C., DeMott, P. J., Hill, T. C. J., and Huffman, J. A.: Bioaerosol field measurements: Challenges and perspectives in outdoor studies, Aerosol Science and Technology, 1-27, 10.1080/02786826.2019.1676395, 2019.)
We ask the authors to change this paragraph.

---

## Author Response (AR2)

**Manuscript format description:**

Black text shows the original referee comment, blue text shows the authors response, and red text shows quoted manuscript text. Changes to the manuscript text are shown as *italicized and underlined*. We used bracketed comment numbers for referee comments (e.g., [R1.1]) and author's responses (e.g., [A1.1]). Line numbers refer to the discussion/review manuscript.

**Anonymous Referee #2**

Received: 27 April 2021

[R2.1] We thank the authors for their effort to answer the concerns of the various reviewers and for the corrections of the manuscript. I agree it could be published but I would like the authors to make a few changes to take into account my previous comments.

[A2.1] We appreciate the constructive comments by Referee #2, which have been considered and implemented in the course of this second revision. The referee's comments and our responses are outlined in detail below:

**Minor comments:**

[R2.2] P3 Line 36 : "Our results demonstrate that FISH has great potential in bioaerosol analysis as it provides number concentrations of specific organism classes (i.e., from domain down to species level) and, therefore, combines bioaerosol identification, enumeration, an visualization" I suggest to delete "great" and keep "FISH has potential in bioaerosol analysis". Indeed we think that this technique is so heavy that its use to analyze a great number of samples will be limited, alternative techniques will be more suitable (qPCR, sequencing, Flow cytometry + targeted probes etc..)

P 11 line 7: "Our study showed that FISH has great analytical potential in bioaerosol analysis." Please delete "great" (for the same reason as before).

P11 line 11: "Here, we propose FISH to be a promising tool". Please change "promising" for "interesting" (same reason).

[A2.2] We have changed the manuscript as the referee suggested:

(P3, L36-L38): Our results demonstrate that FISH has  potential in bioaerosol analysis as it provides number concentrations of specific organism classes (i.e., from domain down to

species level) and, therefore, combines bioaerosol *identification, enumeration, and visualization.*

(P11, L7): Our study showed that FISH has  analytical potential in bioaerosol analysis.

(P11, L11-12): Here, we propose FISH to be *an interesting* tool, to complement the methods currently established for environmental bioaerosol analysis (Sect. S1.4).

[R2.3] supplement, P 5 line: The authors should add a paragraph about the advantages of flow cytometry to quantify the total number of cells and look at their size distribution. This technique is very fast contrary to DAPI which is time consuming. They should also speak about the combination of flow cytometry with specific staining with targeted probes.

[A2.3] We have added these aspects according to the referee's suggestions in the supplement chapter "Bioaerosol analysis methods" to point out the potential of flow cytometry in relation to FISH/DAPI staining with manual counting. We tried to find the right balance between covering the pros and cons of individual methods, on one hand, and going into too much detail on a single method, on the other, as this is not the focal point of this study. In fact, an appropriate and critical comparison of FISH/DAPI staining, flow cytometry, and qPCR would probably require a dedicated study, which addresses the various aspects to be considered in the choice of the 'right' tool(s) for bioaerosol analysis in a given environment. The referee is right that flow cytometry appears to have clear "advantages" over the manual FISH/DAPI staining and counting approach, which include the fact that it is "very fast" and directly provides particle "size distributions". This portrait would be incomplete, however, without mentioning that flow cytometry (as essentially every technique) has drawbacks at the same time (e.g., uncertainties due to an autofluorescence background, challenging sampling logistics at remote sites, lower precision in quantification).

We agree that flow cytometry works out best in combination with fluorescence staining for environmental bioaerosol analysis. This reduces the influence of autofluorescence and thus the detection of interfering non-biological particles as described in the supplement section S1.4. The occurrence autofluorescence from biological and non-biological materials across wide intensity and wavelength ranges has been well document before and represents a prominent challenge in automated bioaerosol detection (Pöhlker et al., 2012; Savage et al., 2017; Huffman et al., 2020). DNA or RNA staining creates fluorescence in a rather narrow spectral range and, thus, enables a targeted detection via e.g. microscopy counting or flow cytometry. The following text has been added to the supplement.

P6, L3- P6, L 9: *The microscopic analysis of FISH treated samples also bears drawbacks such as the time consuming manual enumeration of fluorescent single particles. Here, automated image generation and software based particle detection or sample analysis with flow cytometry could improve the analysis by speeding up the process. The application of these two techniques is dependent on the careful characterization of sample's properties such as aerosol mixing state and diversity, sample purity or abundance of interfering materials.*

[R2.4] supplement, P5 line 23. We do not agree with this paragraph :impingement is recognized as an efficient tool and no growth is observed with a short time collection, typically less than an hour is need to have enough sample , especially using high volume impingers (Šantl-Temkiv, T., Sikoparija, B., Maki, T., Carotenuto, F., Amato, P., Yao, M., Morris, C. E., Schnell, R., Jaenicke, R., Pöhlker, C., DeMott, P. J., Hill, T. C. J., and Huffman, J. A.: Bioaerosol field measurements: Challenges and perspectives in outdoor studies, Aerosol Science and Technology, 1-27, 10.1080/02786826.2019.1676395, 2019.)

We ask the authors to change this paragraph.

[A2.5] We thank reviewer 2 for this comment. As written in the text, the experimental issues play a role when liquid evaporation over time takes place because of long sampling periods. Since we purposefully collected diel average samples in this study, liquid evaporation became a relevant issue here. We modified the following sentence to point out this aspects more clearly. (Supplement, P5, L24- P6, L 3): However, *for long-term sample collections as conducted for this study* varying collection efficiency due to liquid evaporation over time and therefore changes in chemical composition (e.g., pH or fixative concentration) as well as microbial growth within the liquid *can play a role* and have to be considered (Lin et al., 1997) .

Further note that we do not question that liquid impinges are an efficient bioaerosol sampling device in a range of applications. There are multiple studies analyzing their performances and proofing their advantages (e.g.: Lin et al, 1997; Dybwad et al., 2014; Šantl-Temkiv et al., 2017). However, we found that dry aerosol filtration was more appropriate than impingement for this study in the Amazon for the reasons given in the supplement. In fact, we conducted the first sampling for FISH in the Amazon with the so called BioSampler, which is a broadly used impinger (Willeke et al., 1998; Lin et al., 2010), and compared its performance with filter sampling. We found that for the relevant sampling period varying liquid levels and therefore a varying collection efficiency and varying concentration of the fixative occurred. Furthermore, microbial growth has been an omnipresent challenge in the Amazon, as the high humidity and

abundant airborne fungal and bacterial spores foster rapid microbial growth. Everything, even the laboratory equipment, was affected by microbial, especially fungal, growth on its' surface, if it was not disinfected and cleaned very often. As a result, if there is no or not enough fixative in the sampling medium, there indeed is a high potential for microbial growth in the sampler.

**References**

Dybwad, M., Skogan, G., and Blatny, J. M.: Comparative Testing and Evaluation of Nine Different Air Samplers: End-to-End Sampling Efficiencies as Specific Performance Measurements for Bioaerosol Applications, Aerosol Science and Technology, 48, 282-295, 10.1080/02786826.2013.871501, 2014.

Huffman, J. A., Perring, A. E., Savage, N. J., Clot, B., Crouzy, B., Tummon, F., Shoshanim, O., Damit, B., Schneider, J., Sivaprakasam, V., Zawadowicz, M. A., Crawford, I., Gallagher, M., Topping, D., Doughty, D. C., Hill, S. C., and Pan, Y.: Real-time sensing of bioaerosols: Review and current perspectives, Aerosol Science and Technology, 54, 465-495, 10.1080/02786826.2019.1664724, 2020.

Lin, X., Willeke, K., Ulevicius, V., and Grinshpun, S. A.: Effect of sampling time on the collection efficiency of all-glass impingers, American Industrial Hygiene Association Journal, 58, 480-488, 1997.

Lin, X., Reponen, T., Willeke, K., Wang, Z., Grinshpun, S. A., and Trunov, M.: Survival of Airborne Microorganisms During Swirling Aerosol Collection, Aerosol Science and Technology, 32, 184-196, 10.1080/027868200303722, 2000.

Pöhlker, C., Huffman, J. A., and Pöschl, U.: Autofluorescence of atmospheric bioaerosols - fluorescent biomolecules and potential interferences, Atmos. Meas. Tech., 5, 37-71, 10.5194/amt-5-37-2012, 2012.

Šantl-Temkiv, T., Amato, P., Gosewinkel, U., Thyrhaug, R., Charton, A., Chicot, B., Finster, K., Bratbak, G., and Löndahl, J.: High-Flow-Rate Impinger for the Study of Concentration, Viability, Metabolic Activity, and Ice-Nucleation Activity of Airborne Bacteria, Environmental Science & Technology, 51, 11224-11234, 10.1021/acs.est.7b01480, 2017.

Savage, N. J., Krentz, C. E., Könemann, T., Han, T. T., Mainelis, G., Pöhlker, C., and Huffman, J. A.: Systematic characterization and fluorescence threshold strategies for the wideband

integrated bioaerosol sensor (WIBS) using size-resolved biological and interfering particles, Atmos. Meas. Tech., 10, 4279-4302, 10.5194/amt-10-4279-2017, 2017.

Willeke, K., Lin, X., and Grinshpun, S. A.: Improved aerosol collection by combined impaction and centrifugal motion, Aerosol Science and Technology, 28, 439-456, 1998.